

# Why does the signal-to-noise paradox exist in seasonal climate predictability?

Yashas Shivamurthy[1,2], Subodh Kumar Saha[1], Samir Pokhrel[1], Mahen Konwar[1], and Utkarsh Verma[1,2]

[1]Indian Institute of Tropical Meteorology, Pune, India
[2]Savitribai Phule Pune University, Pune, India

**Correspondence:** Subodh Kumar Saha (subodh@tropmet.res.in)

**Abstract.** Estimates of the potential predictability limit (PPL) for seasonal climate, typically based on a perfect model framework, sometimes encounter challenges of being paradoxical, as actual skill surpasses the PPL. The signal-to-noise paradox (SNP) gets its name from the use of model-based signal-to-noise ratios to estimate the PPL. Here, we study seasonal climate predictability in the tropical and subtropical regions during the boreal summer (June to September), with a focus on the SNP. We estimate PPL within the perfect model framework, only considering error growth from initial conditions. Signal and noise components display temporal non-orthogonality and a weak association between estimates of PPL and actual prediction skill, contradicting its intended purpose. Moreover, paradoxical regions do not align with significant correlations between signal and noise, indicating that the accurate separation of seasonal forecasts into signal and noise components alone is not sufficient to avoid paradoxes. We have also demonstrated that sub-seasonal components, which are building blocks of seasonal mean, substantially contribute to seasonal anomalies in association with major global predictors. The co-variability between sub-seasonal components and seasonal anomalies is wide-ranging and often skewed compared to observations, thereby influencing seasonal prediction skills and PPL. Therefore, a robust PPL estimation should consider errors from initial conditions and model-related factors such as physics, dynamics, and numerical methods. In this context, we propose a novel method to estimate the PPL of seasonal climate, which can be free from paradoxical situations.

## 1 Introduction

Seasonal climate prediction plays a pivotal role in making policy decisions, facilitating long-term planning, and implementing mitigation strategies, ultimately contributing to the development of a climate-resilient society. The emergence of the first computer in the 1950s initially fostered optimism for achieving precise weather and climate prediction. However, studies by Lorenz (1963, 1969) pointed out the inherent unpredictability of weather beyond a limited temporal horizon of only a few days. Consequently, this revelation led to the perception that the endeavour of seasonal prediction was considered unattainable. Nevertheless, a significant advancement in our understanding of long-lead predictability in the tropical climate was achieved by Charney and Shukla (1981), with further elaboration by Shukla (1998). In this context, the predictability of seasonal climate is attributed to the influence of slowly evolving boundary conditions, such as sea surface temperature (SST) and soil mois-



ture, which retain memory and significantly influence atmospheric instabilities on longer timescales. This breakthrough has established the scientific foundation for seasonal climate prediction.

While significant advancements have been made in improving the accuracy of weather forecasts over the past few decades (e.g., Bauer et al., 2015), the task of providing reliable seasonal predictions remains a formidable challenge. The Indian summer monsoon serves as an illustrative example, as attempts to predict it date back a century (Walker, 1924; Mooley and

Parthasarathy, 1984), yet achieving a consistently reliable prediction remains elusive (Shukla, 2007; Jain et al., 2019). In addition to evaluating current prediction skills, it is of equal importance to ascertain the inherent constraints of seasonal climate predictability, denoted as the Potential Predictability Limit (PPL). The PPL serves as an upper threshold for a model's prediction skill, beyond which further enhancements become unattainable. Conventional wisdom makes us believe that a dynamical prediction system cannot surpass the PPL (Kumar et al., 2005; Rajeevan et al., 2012). However, several studies have docu-

mented instances where a model's seasonal prediction skill surpasses the estimated PPL (e.g., Kumar et al., 2014; Scaife et al., 2014; Saha et al., 2016a; Scaife and Smith, 2018), giving rise to a signal-to-noise paradox (SNP), a situation where the proportion of predictable variance in models is too weak. This translates into actual skill (correlation skill between ensemble mean and observation) being greater than the estimated PPL.

In previous studies, attempts to estimate the potential predictability of seasonal climate were predominantly focused on

shorter hindcasts and employed various estimation methods (Yang et al., 2012; Saha et al., 2016b, a, 2019; Scaife et al., 2014). However, recent studies have highlighted the pervasive challenge of the SNP, which extends across different climate models and temporal scales (Scaife and Smith, 2018; Strommen and Palmer, 2019; Zhang and Kirtman, 2019; Zhang et al., 2021; Sévellec and Drijfhout, 2019). Intriguingly, about 70 percent of the globe exhibits SNP in annual surface air temperature simulated by CMIP5 models (Sévellec and Drijfhout, 2019). This paradox is not solely attributed to issues with initialization

processes (Zhang and Kirtman, 2019; Cottrell et al., 2024). Addressing the low signal-to-noise ratio present in models has been recognized as a potential solution to improve model future projections (Smith et al., 2020). Model's response to external radiative forcing and limited ensemble size may also be responsible for SNP (Klavans et al., 2021). Nonetheless, another study utilizing the Lorenz 1963 model (Lorenz, 1963) argues that the paradox arises from the initialization process, wherein the initial ensemble spread (standard deviation) surpasses the observational spread (Mayer et al., 2021). Large ensemble sizes

and the application of post-processing techniques also hold promise for enhancing prediction accuracy by reducing noise and enhancing forecast variance (Eade et al., 2014; Smith et al., 2019). Hu et al. (2021) have shown that in NCEP CFSv2, the spatial pattern of the primary mode of signal and noise components for SST coincides over the tropical Pacific, yet their temporal variability remains uncorrelated. Moreover, the non-stationary nature of the climate system and smaller sample size (i.e. ensemble member and number of years) can confound the detection of the paradox (Weisheimer et al., 2018).

This study aims to explore the SNP in the context of seasonal prediction skill and sub-seasonal variability within tropical and subtropical regions, with a specific emphasis on South Asia and the central Pacific from June to September (JJAS). We estimate the PPL using traditional methods and assess correlation skills for rainfall, surface temperature, sea surface temperature, and mean sea level pressure (MSLP). Subsequently, we apply these methods to identify regions where the paradox exists. While the estimation methods of PPL are rooted in the idealized notion of a perfect model framework, where the model itself is





considered flawless, but errors in the initial conditions result in divergence (i.e., simulations deviate with minor perturbations in initial conditions), it is essential to acknowledge that dynamical models are inherently imperfect, and the realization of a perfect model is improbable. Furthermore, estimates of PPLs are often very close to actual forecast skill and are expected to increase with advancements in the model. This raises critical questions regarding the overall usefulness of PPL. In this context, we posit that the signal and noise components in seasonal mean are not solely attributed to errors in the initial conditions but also

to inadequate representation of physical processes and their associated non-linear feedback mechanisms within a dynamical model. We demonstrate a substantial contribution of sub-seasonal components to the overall variability and predictability of seasonal climate. Additionally, we identify a systematic error in the co-variability between sub-seasonal components and seasonal anomalies, which affects the accuracy of seasonal climate predictions and predictability. Here, we propose a new method for estimating PPL that is closely related to actual skill and potentially free from paradoxical behaviour. This method

serves to determine the upper bound on seasonal climate prediction skill attainable by a model given specific initial conditions. In consideration of the constraint posed by a finite sample size (i.e. 52 ensembles for 41 years) and its potential impact on the accuracy of potential predictability estimates, we employ the Bootstrapping method. This statistical technique enables us to infer the PPL of the entire ensemble population by re-sampling from the available ensembles, thereby mitigating the influence of sample size limitations on our estimates. Detailed data and methods are provided in section 2, while section 3 presents the

study's results, and section 4 summarizes the findings.

## 2 Data and Methods

### 2.1 Model and Experiments

The modified version of the Climate Forecast System version 2 (CFSv2; Saha et al., 2014a, b; Hazra et al., 2017) of the National Centers for Environmental Prediction (NCEP) is used in this study. The CFSv2 consists of a spectral atmospheric model (i.e.

Global Forecast System) at a horizontal resolution of about $1°$ (i.e. T126) with 64 hybrid vertical levels and the GFDL Modular Ocean Model version 4p0d (Griffies et al., 2004) with 40 vertical layers and $0.25 - 0.50°$ horizontal grid spacing. The CFSv2 is coupled with a two-layer sea ice model (Wu et al., 1997; Winton, 2000) and land surface model Noah with four layers of soil and single layer snow scheme (Ek et al., 2003).

      As CFSv2 shows a maximum (minimum) ISMR skill with February and April (March and May) initial conditions (e.g., Saha

et al., 2016a; Pokhrel et al., 2016), based on NCEP's Climate Forecast System Reanalysis (CFSR; Saha et al., 2010), February initial conditions are used to generate ensemble re-forecasts/hindcasts for the years 1981–2021. The model is initialized every day (15 February to 27 February) and four cycles in a day (00, 06, 12, and 18 GMT). Therefore, 52 ensemble member simulations are performed, each year spanning 9-months.



## 2.2 Observed Data

In this study, we employ multiple observed/reanalysis data sets for analysis and comparisons with model simulations. Over land, 2m air temperature data from the Climatic Research Unit (CRU TS3.1; Harris et al. (2014)) is used. SST data is taken from EN4 reanalysis (Good et al., 2013). MSLP data is obtained from ERA5 reanalysis (Hersbach et al., 2020). Daily rainfall data from the Global Precipitation Climatology Project (GPCP Version 1.3, $1° \times 1°$; Huffman et al., 2001) for 1997-2021 and monthly rainfall data from GPCP version 2.3 (Adler et al., 2020) for 1979-2021 with $2.5° \times 2.5°$ horizontal resolution are employed.

## 2.3 Methodology

Here, we employ the Analysis of Variance (ANOVA) method based on the perfect model framework (e.g., Rowell et al., 1995; Rowell, 1998; Schneider and Griffies, 1999) to estimate PPL. The PPL serves as a measure of the model's ability to predict the Earth's weather and climate with the utmost skill, where prediction skill should always be lower than the PPL due to unavoidable errors in the initial conditions. However, it is found that the prediction skill of a model is higher than the PPL on several occasions and across the globe, which is termed the signal-to-noise paradox (SNP). Furthermore, to verify the existence of SNP, the Ratio of Predictable Components (RPC) is used (Eade et al., 2014).

### 2.3.1 ANalysis Of VAriance (ANOVA) method

In this method, the total variance is split into signal and noise components, i.e. external ($\sigma^2_{EXV}$) and internal ($\sigma^2_{IV}$) variances, respectively. Here, predictable and unpredictable components are termed external/signal and internal/noise components, respectively. The ratio of external to internal variance is known as the signal-to-noise ratio (SNR). If x is the precipitation field of the model, i is the year of the model integration (total year 'N'), and j is the number of ensemble simulations (total ensemble n = 52), then internal variance following Rowell et al. (1995), can be expressed as

$$\sigma^2_{IV} = \frac{1}{N(n-1)} \sum_{j=1}^{n} \sum_{i=1}^{N} (x_{ij} - \overline{x_i})^2 \tag{1}$$

where $\overline{x_i} = \frac{1}{n} \sum_{j=1}^{n} x_{ij}$ is the ensemble mean of the model for a year and the degrees of freedom is N(n – 1). The variance of ensemble mean ($\sigma^2_{EV}$) can be estimated as

$$\sigma^2_{EV} = \frac{1}{(N-1)} \sum_{i=1}^{N} (\overline{x_i} - \overline{\overline{x}})^2 \tag{2}$$

where $\overline{\overline{x}} = \frac{1}{Nn} \sum_{j=1}^{n} \sum_{i=1}^{N} x_{ij}$ is the average over all year and all ensemble. However, the variance of the ensemble mean is a biased estimate of external variance (Scheffe, 1959). As the number of ensemble members is not very large (here 52), the ensemble mean contains residual internal variability. Therefore, the external variance may be estimated following Scheffe





(1959) as

$$\sigma_{EXV}^2 = \sigma_{EV}^2 - \frac{1}{n}\sigma_{IV}^2 \tag{3}$$

Therefore, the total variance can be estimated as

$$\sigma_{TV}^2 = \sigma_{EXV}^2 + \sigma_{IV}^2 \tag{4}$$

The external variance arises due to slowly varying quasi-periodic or aperiodic boundary conditions, such as El-Niño Southern Oscillation (ENSO), which evolve on a longer time scale than the predictand (e.g. seasonal monsoon rainfall). In general, the power associated with quasi-periodic slowly evolving processes is higher than that of faster-evolving processes (Peixoto and Oort, 1992, , Figure 2.7). Hence, the slowly evolving process may dictate the evolution of the faster processes, contributing to predictability. The ratio of external to internal variance (a quantitative measure of predictability) is known as the signal-to-noise

ratio (or SNR). A perfect coupled model does not attest to a perfect seasonal rainfall forecast due to unavoidable errors in the initial conditions. Therefore, there will always be an upper limit to the seasonal predictability (e.g., Kang and Shukla, 2006; Westra and Sharma, 2010), which can be expressed in terms of SNR as

$$PPL_{ANOVA} = \sqrt{\frac{SNR}{SNR+1}} \tag{5}$$

The values of PPL vary between 0 and 1 and show the maximum limit of correlation skill achievable by a model at a given

lead time.

### 2.3.2    Ratio of Predictable Components (RPC)

RPC method (Eade et al., 2014) is used to identify the signal-to-noise paradox. The RPC is the ratio of predictable components in observation to that in the model. The predictable component in observation (PC$_{obs}$) is estimated directly from the fraction of the variance that can be explained by model forecasts (i.e. correlation). The predictable component in the model (PC$_{model}$)

is estimated by the ratio of the variance of the ensemble mean to the variance of individual ensemble members. Ideally, RPC should be 1, as the observation and model should contain the same proportion of predictable variance, and the squared correlation should match the predictable proportion of variance in the model. Regions where the RPC index is greater than 1, show that the predictable component in the model is less than that we see in the real world (i.e. paradox). The RPC is given by

$$RPC = \frac{PC_{obs}}{PC_{mod}} \geq \frac{r}{\sqrt{\sigma_{IV}^2/\sigma_{TV}^2}} \tag{6}$$

### 2.3.3    Projection of sub-seasonal variance on seasonal anomaly

Parameters like precipitation are not continuous phenomena; they occur as discrete events (either zero or some positive value). The sub-seasonal components (e.g., precipitation events on hourly, synoptic, or MISOs timescales) are the building blocks of





the seasonal or annual mean. The seasonal mean precipitation is simply the sum of precipitation during these events. Moreover, various sub-seasonal bands contributed differently to the seasonal mean. As the sum of these sub-seasonal events forms the seasonal mean, year-to-year variability in the characteristics of sub-seasonal components also contributes to seasonal anomalies. In other words, global predictors affect seasonal precipitation by modulating sub-seasonal components, such as changing the intensity and duration of these events (e.g. Saha et al., 2019, 2020, 2021).

Due to imperfections in model physics, accurately simulating sub-seasonal components remains a challenge. Therefore, their anomalous contributions to seasonal anomalies are problematic. For example, in the Indian summer monsoon, synoptic systems (i.e., lows and depressions) contribute about 45–55% of the seasonal mean precipitation (e.g. Yoon and Chen, 2005), and account for the maximum year-to-year variability (Saha et al., 2020). Therefore, errors in the co-variability between seasonal mean and sub-seasonal variance can also contribute to ensemble spread. To demonstrate this, we analyzed the seasonal variance of prominent sub-seasonal bands in the rainfall time series in relation to the seasonal anomaly. The seasonal variance of these sub-seasonal components serves as a measure of their vigour or strength in a season. It is important to note that errors in simulating precipitation arise from imperfections in various physical processes and their interactions. We chose precipitation to demonstrate the role of model physics in ensemble spread because it is a highly sought-after forecast parameter for society and, at the same time, has significant uncertainty. The time series of daily rainfall of a year (area average or a single point) can be represented by the following equation

$$x_T = x_c + x_a + \sum_f x_f \tag{7}$$

where, $x_T$ is the total rain, $x_c$ is the climatological mean annual cycle, $x_a$ is the external signal or anomalous annual cycle, $x_f$ represents the internal components consisting of all subseasonal frequencies ($f$). Using harmonic analysis, the sum of the mean and the first three harmonics represents the 'smooth annual cycle' in the daily time series for a year. Here, $x_c$ is the climatological mean of the 'smooth annual cycle', and $x_a$ is the deviation of the 'smooth annual cycle' of a year from the climatological mean annual cycle. Therefore, after re-arrangement, the above equation can be written as

$$(x_T - x_c) = x_a + \sum_f x_f \tag{8}$$

The left-hand term represents the total daily anomaly. In terms of seasonal variance, using daily June-to-September data (122 days) equation 8 for a particular season can be written as

$$\sum_{l=1}^{122}(x_T^l - x_c^l)^2 = \sum_{l=1}^{122}(x_a^l)^2 + \sum_{f=1}^{K}\sum_{l=1}^{122}(2x_a^l \cdot x_f^l) + \sum_{f=1}^{K}\sum_{l=1}^{122}(x_f^l)^2 \tag{9}$$

$$V_T = V_E + \sum_f (V_{cov}) + \sum_f (V_{IV}) \tag{10}$$





where $l$ represents the day, $V_T$ is the total variance, $V_E$ is the external variance, $V_{cov}$ is the covariance between internal and external components, $V_{IV}$ represents the internal variance, $K$ is the number of sub-seasonal bands (e.g., synoptic, bi-weekly) in a season. However, due to orthogonality, the covariance term becomes negligible. In terms of seasonal anomaly, equation 10 can be written as

$$V_T' = V_E' + \sum_f V_{IV}' \tag{11}$$

Where $V_T'$, $V_E'$ and $V_{IV}'$ are seasonal anomalies of the total variance, external variance and internal variance, respectively. Let $I'$ be the seasonal-mean anomaly then, the covariance between the seasonal-mean anomaly and seasonal-variance anomaly can be written as

$$\sum_i V_T' I' = \sum_i V_E' I' + \sum_i \sum_f V_{IV}' I' \tag{12}$$

The left-hand term of equation 12 represents the interannual covariance between total sub-seasonal variance and seasonal mean. The first term on the right-hand side represents the covariance between external variance (i.e., the signal component) and the seasonal mean, while the second term represents the covariance between internal variance and the seasonal mean. It is important to note that the first term on the right-hand side explicitly does not contain information on the building blocks of the seasonal mean and is, therefore, not used in our analysis. On the other hand, the last term is of particular interest, as it represents the interannual covariance between seasonal mean and sub-seasonal bands. Since sub-seasonal components are considered the building blocks of the seasonal mean/anomaly, analyzing the covariance within sub-seasonal bands may provide insights into their contributions and the causes of model bias. Parameters like rainfall do not occur continuously but as discrete events (i.e., sub-seasonal components). The sum of these events constitutes the seasonal rainfall. Therefore, a global predictor (e.g., ENSO, AMO) must modulate the sub-seasonal components to affect the seasonal rainfall (e.g. Saha et al., 2019; Borah et al., 2020). For a similar reason, variability and predictability of the seasonal mean monsoon rainfall may be affected due to imperfect physics, dynamics and numerical schemes through error in the simulation of sub-seasonal components. Fourier analysis is used to construct a smooth annual cycle and total anomaly, while a Lanczos band-pass filter (Duchon, 1979) is applied to filter daily time series data into synoptic (2-5 days), super-synoptic (10-20 days), and Monsoon Intra-Seasonal Oscillations/Madden-Julian Oscillation(MISO/MJO) (20-60 days) bands.

## 3 Results

### 3.1 Regions of paradox

The assessment of PPL derived from ANOVA (equation 5) and the existence of SNP, which is also confirmed by RPC (equation 6), is the central focus of this study. This comprehensive analysis is conducted on seasonal (June-July-August-September)





averaged rainfall, SST, MSLP, and 2-metre air temperature (land region) data of 1981-2021, as illustrated in Figure 1. To find out the potential presence of an SNP, we subtract correlation skill from the PPL estimated by ANOVA. Our analysis reveals a conspicuous pattern across a substantial portion of the global tropical and sub-tropical regions, wherein the correlation skill surpasses the estimated PPL (including the Indian region for rainfall and mean sea level pressure), thus manifesting a paradoxical situation (stippled regions with black dots in Figure 1). Notably, these tropical regions exhibit a higher degree of predictability, with the notable exception of rainfall, which consistently demonstrates the lowest level of predictability among the considered parameters. The existence of SNP is further confirmed through RPC, represented by semi-transparent white shading that predominantly overlaps the paradox regions identified by the ANOVA method. It is worth emphasizing that this intriguing SNP in seasonal prediction has been recognized in several previous studies as well, including works by Scaife et al. (2014); Kumar et al. (2014); Eade et al. (2014); Scaife and Smith (2018); Saha et al. (2019). However, the reasons for the paradoxical behaviour in the seasonal forecast are not clear (e.g. Scaife and Smith, 2018).

Existing studies suggest that the SNP predominantly arises in the context of seasonal forecasting, which encompasses timescales of a month and beyond, as opposed to weather or medium-range predictions. This distinction can be attributed to the fundamental nature of weather forecasting, characterized as an initial value problem, where precision in the initial conditions takes precedence. Conversely, when dealing with forecasts spanning from seasonal to decadal and even longer timescales, slowly evolving boundary conditions (e.g., SST, soil moisture) and external forcings (e.g., solar variability, volcanic eruptions) become more prominent. These factors are the sources of predictability of the second kind. As the forecast lead time increases, the influence of initial errors wanes. In essence, in the realm of climate prediction, forecasts made several months or decades ahead depend significantly on the accurate representation of external drivers (e.g., greenhouse gas concentrations, aerosol emissions), the model's dynamics, its underlying physics, and the feedback mechanisms that operate within the model (i.e., internal variability). It's worth noting that the methodology employed in this study, namely ANOVA, is based on the framework of the 'perfect model' paradigm. Here, the model is considered perfect or flawless, and any deviations in model simulations are attributed solely to errors in the initial conditions. This principle becomes particularly evident in large ensemble simulations, where even slight differences in initial conditions result in divergent model outcomes.

As estimates of PPL and RPC depend on the signal-to-noise ratio (section 2.3), the time series of the seasonal mean is partitioned into external variability (i.e., signal component) and internal variability (i.e., noise). For an accurate estimate of PPL, signal and noise components should be clearly separated, i.e., both should be independent of each other. Therefore, the adequacy of partitioning model re-forecasts into signal and noise components by the ANOVA/RPC method comes under scrutiny. Given that signal and noise inherently exhibit orthogonality, we examine the correlation between these components, aiming to ascertain the robustness of their partitioning. For rainfall, a substantial region in the tropics and sub-tropics exhibits statistically significant correlations between signal and noise components (Figure 2). Conversely, in the case of MSLP and SST/LST, the significant area is smaller but extends across tropical and sub-tropical regions. Weisheimer et al. (2018) hypothesized that paradox emerges due to statistical problems in estimating RPC. Bröcker et al. (2023) also highlights this point, that the Anomalous Signal-to-Noise Ratio (ASNR, i.e. RPC >1) is not purely noise-based but contains a systematic error component. Importantly, it should be noted that the regions displaying the paradox do not necessarily align with those demonstrating high or significant





**Figure 1.** Potential predictability (shading) based on ANOVA method using equation no. 5. for JJAS averaged a) rainfall, b) mean Sea level pressure, c) Sea/land surface temperature using CFSv2 re-forecast of 41 years (1981-2021) and 52 ensemble members. Signal-to-noise paradox regions (where model correlation skill with observations is higher than the potential predictability) are stippled with black dots. White semi-transparent regions, mostly coinciding with stippled regions, represent $RPC > 1.0$.



**Figure 2.** Test of orthogonality between signal and noise components. Correlations between internal (i.e., noise) and external (i.e., signal) components of JJAS averaged a) rainfall, b) mean Sea level pressure, c) Sea/land surface temperature employing CFSv2 re-forecast of 41 years (1981-2021) and 52 ensemble members. Correlations significant at 95% (Student's two-tailed t-test) and above are stippled by black dots.





correlations between signal and noise. Hence, it is plausible that precise partitioning into signal and noise components alone may not be sufficient to eliminate paradoxical behaviour and, consequently, determine the true limit of potential predictability.

## 3.2 Signal-to-Noise Paradox in Relation to Model Skill

As the name suggests, PPL is the maximum achievable skill by a model. However, its estimate does not have any binding relationship with the observations (i.e. observations are not required to estimate PPL). Nevertheless, it is important to understand how strong is the association between actual skill and estimated PPL. For that, we use time series data involving a predictor (Niño3.4 SST) and rainfall within two delineated regions (as shown in Figure 1): one situated in the paradoxical region, denoted as the Indian Summer Monsoon Region (ISMR), and the other in a non-paradoxical region, referred to as the Pacific Region (PACR). These two contrasting regions are selected to investigate whether there are distinct variations in PPL with actual skill. Furthermore, to test the association between PPL and actual skill, the ensemble members are arranged in ascending order in their absolute error in seasonal anomaly (with respect to GPCP data) for each year (i.e. ensemble members are arranged in a good-to-poor correlation skill order). We calculate predictability measures and correlations over a 21-ensemble member running window to see the general pattern. In the case of ISMR, although the PPL slightly decreases with diminishing skill, good ensemble members exhibit paradoxical behaviour while poor ensemble members do not (Figure 3). However, RPC varies with actual skill, as it is inherently a function of the correlation between model predictions and observations. For Niño3.4 and PACR, PPL does not fluctuate with actual skill, and the actual skill consistently remains significantly lower than the PPL. This implies that PPL is not contingent on a model's predictive skill. In other words, a higher PPL does not necessarily indicate that a model has the potential to achieve greater skill, which contradicts the objective of estimating PPL. Ensemble members are arranged from best to worst (or good to poor) to clearly show the relationship between actual skill and PPL.

The estimated PPLs clearly do not represent the upper limit of seasonal prediction skill, raising the question of why such a discrepancy exists. A plausible explanation is that partitioning of seasonal mean into signal and noise components is problematic. It assumes, perhaps unrealistically, that the inter-ensemble spread in a year is only due to initial errors (see section 2.3). Subsequently, we compute correlations between signal and noise components using a 21-ensemble moving window of arranged ensembles (Figure 4). Remarkably, the results reveal a distinct pattern where correlations strengthen and become statistically significant for ensemble members exhibiting greater bias in the seasonal anomaly. While the correlation between signal and noise becomes stronger (inverse correlation) as we move from good to poor ensembles (Figure 3), particularly for PACR, the PPL remains relatively constant (Figure 3c). We also note that for the ISMR, even though the paradox persists in the case of good ensemble members (as shown in Figure 3a), the correlation between the signal and noise components is relatively stronger for good (r=0.2) and poor (r=-0.2) ensemble members (Figure 4). Therefore, above all suggest that accurate partitioning into signal and noise components is also not a sufficient criterion for obtaining the true estimates of PPL.

Apart from PPL, it is also important to assess how skillful the model is in predicting both the predictors and the predictands. Given two contrasting regions for analysis of predictands (i.e. ISMR, PACR), we have evaluated the prediction skill for these predictands and major global predictors (i.e. Niño3.4 index, Indian Ocean Dipole (IOD), Pacific Decadal Oscillation (PDO), and Atlantic Multi-decadal Oscillation (AMO)). It is noteworthy that a diverse range of correlation skills is evident among

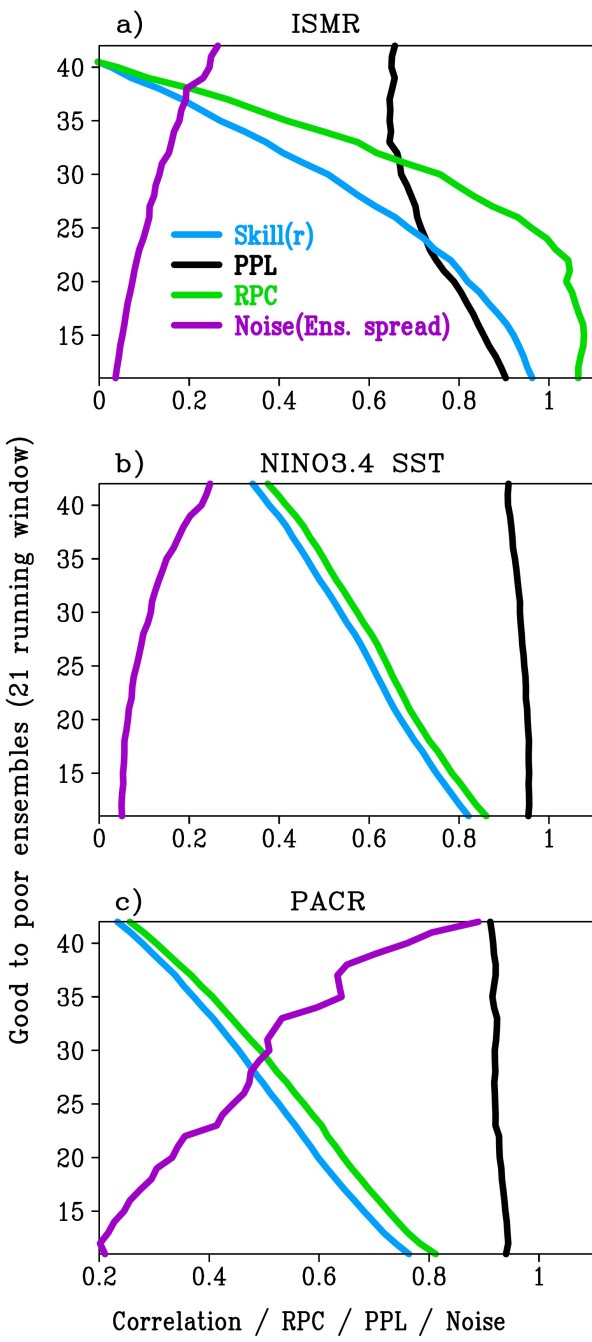

**Figure 3.** Correlation skill (blue), ANOVA based Potential predictability limit (PPL; black), Ratio of Predictable Component (RPC; green) and ensemble spread/'noise' component (purple) of arranged ensembles from good to poor and using a 21-ensemble moving window for a) ISMR rainfall, b) Niño3.4 SST, c) PACR rainfall.



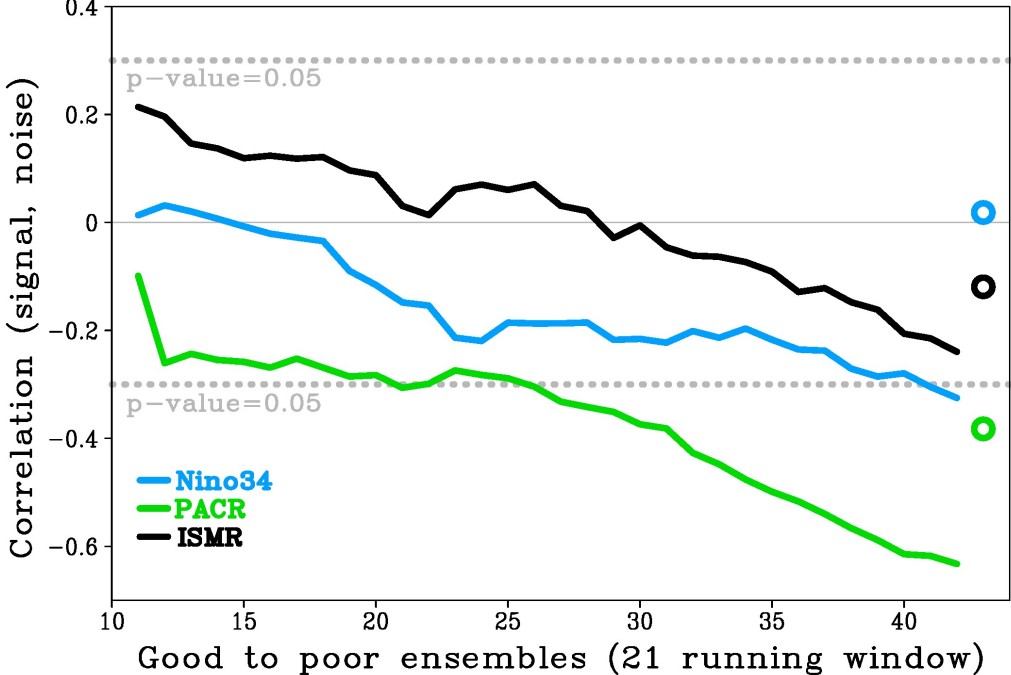

**Figure 4.** Correlations between signal and noise components estimated based on the ANOVA method. Correlations using all 52 ensemble members (open circles) and a 21-ensemble member moving window, where ensemble members are arranged from good to increasingly poor ensembles, are shown for ISMR (black), PACR (green), and Niño3.4 SST (blue). The grey dotted line indicates correlations that are significant at the 95% confidence level.

ensemble members (Figure 5). However, it is important to underscore that the skill exhibited by the ensemble mean (indicated by the red circle) consistently surpasses that of the average of the individual members (depicted by the blue circle). Furthermore, aside from the IOD, the correlation skill for all other predictors and both predictands remains statistically significant at or above

the 99% confidence level. This highlights the model's proficiency in forecasting both the predictors and predictands, which is essential for further analyzing their complex relationships with other components of the climate system, such as sub-seasonal components. However, the RMSE exceeds 100% of the observed standard deviation (Figure 5b), indicating that the model's errors are greater than the natural variability in the observed data.

### 3.3   Role of Sub-seasonal Components

Although the advent of computers in the 1950s gave hope for accurate weather prediction, Lorenz's work (Lorenz, 1963) revealed the inherent unpredictability beyond a few days due to non-linearity in the systems. Subsequently, Charney and Shukla (1981) pioneered the concept of long-lead predictability in the tropics (e.g. monthly/seasonal mean) due to the influence of



slowly varying boundary conditions (e.g., SST, soil moisture) on atmospheric instabilities. Recent studies have shown that sub-seasonal components of the Indian summer monsoon, in association with global predictors (e.g., ENSO, AMO, etc.), sig-
nificantly contribute to the inter-annual variability of the ISMR (Saha et al., 2019, 2020, 2021). The sub-seasonal components are the building blocks of the seasonal mean. Moreover, precipitation is a discrete phenomenon that comes as an event (either zero or a positive value). The annual, seasonal, and diurnal cycles of precipitation are composed of these discrete events. The sum of these events constitutes the diurnal, seasonal, or annual mean. In principle, for any predictor affecting a predictand, the predictor communicates through the modulation of sub-seasonal components, which are then projected onto the season-
al/decadal anomaly of the predictand. However, due to the inherent non-linear characteristics of sub-seasonal components, it is plausible that their variability may not be entirely predictable. In such cases, a portion of the sub-seasonal variability may remain associated with relevant predictors, contributing to overall seasonal predictability (illustrated through schematic in Figure 6). Given that sub-seasonal components constitute fundamental elements of seasonal rainfall, the seasonal prediction skill can be assessed through an examination of the co-variances between sub-seasonal components and seasonal rainfall/predictors
(see section 2.3.3, equations 7-12). Moreover, the co-variance of sub-seasonal components with seasonal rainfall, in relation to PPL and ensemble spread, can likely be attributed to imperfect physics and can be demonstrated. To further elucidate this relationship, we have organized ensemble members in ascending order from more accurate to less accurate, i.e., from good to poor ensembles, based on seasonal rainfall anomaly (ISMR and PACR) compared to observations (i.e. GPCP rainfall).

The sub-seasonal variances are computed within three distinct bands: 2-5 days (synoptic), 10-20 days (bi-weekly/super-
synoptic), and 20-60 days (MISO, MJO). If sub-seasonal components indeed play a role in the prediction skill of seasonal rainfall, their co-variability should exhibit distinguishable patterns among good and poor ensembles. As global daily gridded rainfall observation data (GPCP) is available from October 1997 onward, our analysis is conducted for the period spanning from 1997 to 2021. It is worth noting that similar paradoxical behaviour is evident in the model during this time frame, as depicted in Figure S1. It is evident that the co-variance across these three bands gradually increases from good to poor ensembles in both
the ISM and PAC regions (Figure 7a,b). Notably, for the ISM region, the 2-5 days variability exhibits a distinct contribution to seasonal prediction skill from good to band ensembles, whereas contributions of the 10-20 days and 20-60 days bands are more significant for predicting the seasonal anomalies in the PAC region. Moreover, the model exhibits a considerable spread in the co-variances, particularly in the lower frequency bands (Figure 7c,d). These distributions of co-variances are also skewed in relation to the observations, a phenomenon that carries clear implications for model prediction skill. We also observe that
the noise or ensemble spread increases (Figure 3) as the error in the co-variance grows from good to poor ensemble members (Figure 7a,b).

It may be noted that, in the case of seasonal prediction, the statistics of sub-seasonal components are important and not the actual timing of their occurrence. Moreover, the simulation of sub-seasonal statistics depends on the physics, dynamics, and their coupling with various components of the Earth's climate system (e.g., land, ocean, atmosphere). In general, global
coupled models have serious difficulties in simulating synoptic systems, such as lows and depressions. Nevertheless, these synoptic events contribute significantly to the mean and variability of seasonal climate (e.g. Yoon and Chen, 2005; Saha et al.,




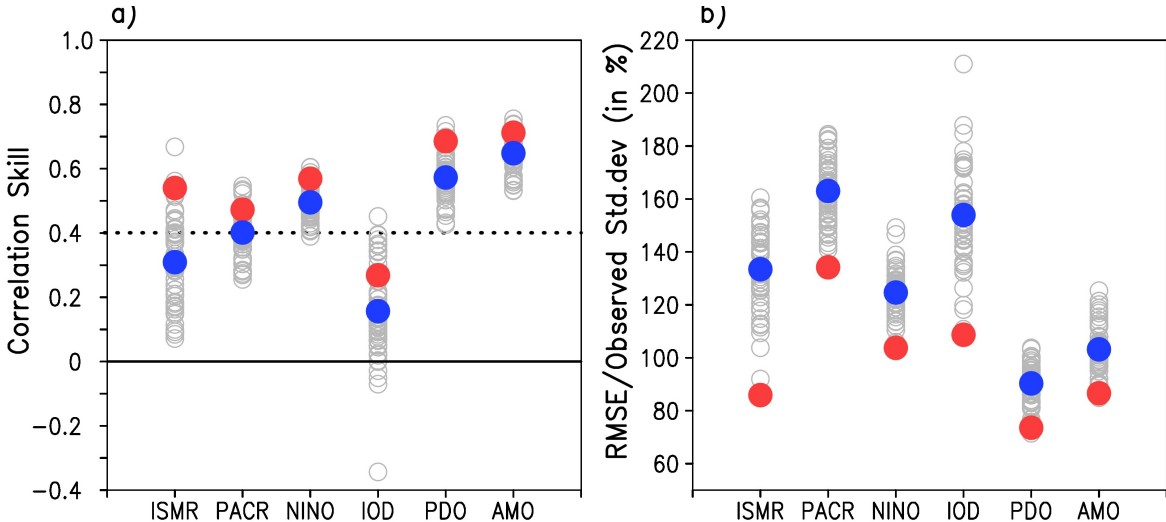

**Figure 5.** a) Prediction skill (correlation) of major predictors (NINO → Nino3.4 index; IOD → Indian Ocean dipole index; PDO → Pacific Decadal Oscillation; AMO → Atlantic Multi-decadal Oscillation) and predictands (ISMR → Indian Summer Monsoon Rainfall averaged over the box shown in Figure 1a; PACR → area-averaged rainfall over the Pacific box (Figure 1a)) in CFSv2 using 41 years of re-forecast (1981-2021) and observations. b) RMSE compared to observed standard deviation (i.e. RMSE/Obs. Std.). Grey open (red solid) circles represent individual (averaged) ensemble member skills. The blue solid circle represents the average skill of individual members (i.e., the average of the grey circles). The black dashed line represents the significance level of correlation values at the 99% confidence level.

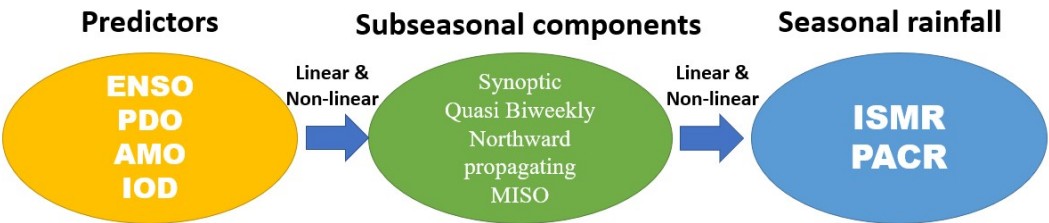

**Figure 6.** A schematic diagram illustrates how slowly varying predictors influence seasonal rainfall anomaly by modulating sub-seasonal components. The sub-seasonal components are considered the building blocks of seasonal rainfall.



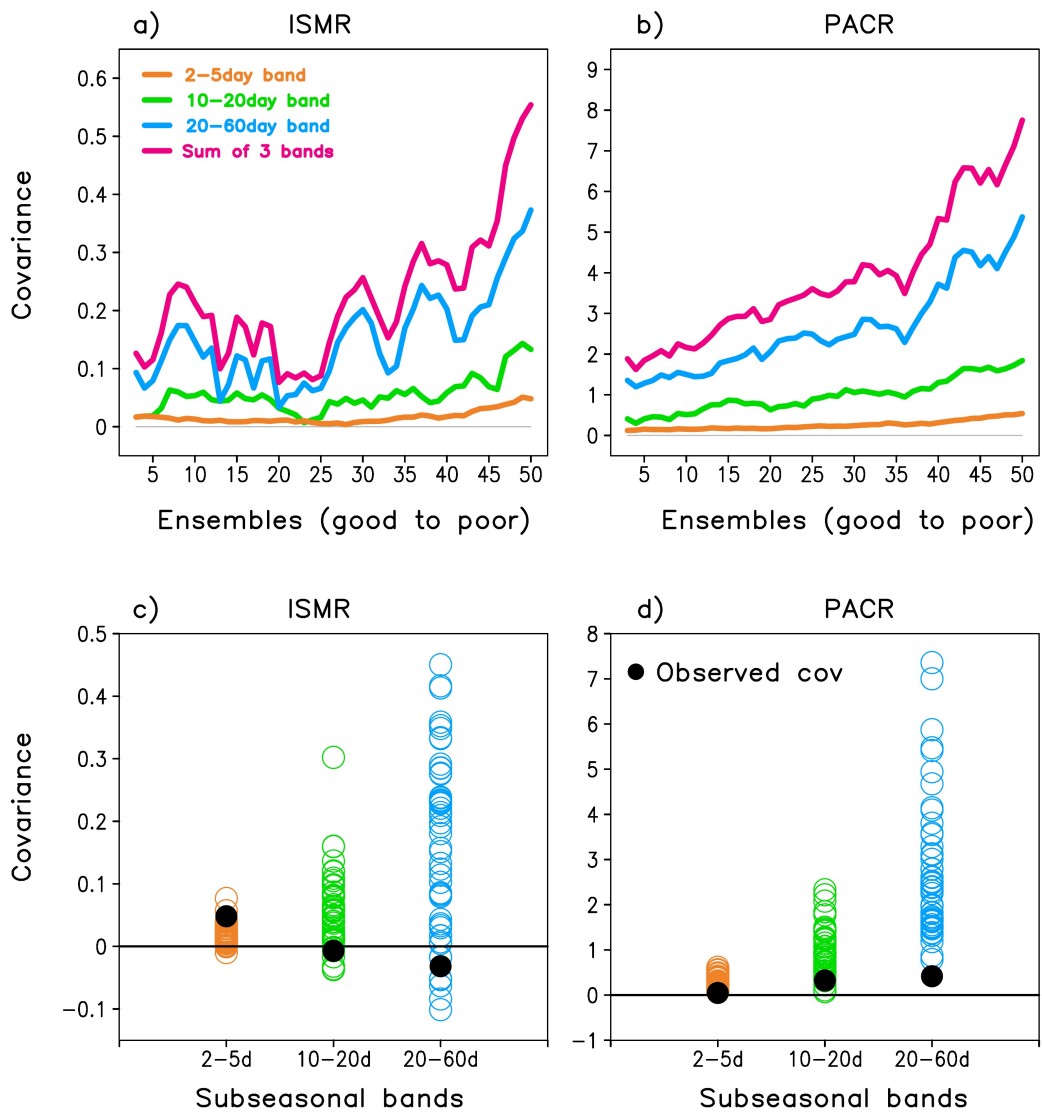

**Figure 7.** Inter-annual co-variance between sub-seasonal variances in three bands (2-5 days, 10-20 days, 20-60 days) and mean rainfall in the ISM and PAC domains (1997-2021). Line plots of co-variances arranged from good-to-poor ensemble members (compared to observed seasonal anomalies) in a 5-ensemble moving average window for a) the Indian summer monsoon domain and b) the Pacific domain. Co-variances in each ensemble member at three sub-seasonal bands, along with observations for c) ISMR and d) PACR.



2019). A disproportional contribution of sub-seasonal components to the seasonal mean is a big concern, as it can spoil the prediction skill, which is evident in Figure 7.

It is now evident that the model's ability to simulate the sub-seasonal contribution to seasonal anomalies significantly in-
fluences seasonal prediction skill. In addition to co-variability, it is equally important that the model accurately replicates the mean sub-seasonal components with high fidelity. Figure 8 provides an overview of the mean sub-seasonal variances within three temporal bands, both in absolute terms and as a percentage of the total variance. The model consistently underestimates synoptic (2-5 day band) variance in both the ISM and PAC regions. In the PAC region, both the 10-20 day and 20-60 day band variances are overestimated. When considering the total variance, the model overestimates it in both regions. However,
when looking at the percentage of these sub-seasonal variances relative to the total variance, the synoptic band is severely underestimated, accounting for only 2-7% of the ensemble member's total variance (Figure 8a,b). It is intriguing to observe that in the observations, the synoptic variance contributes approximately 15% and 26% to the total variance over the ISM and PAC regions, respectively. The 20-60 day band is overestimated in the PAC region while slightly underestimated in the ISM region. Consequently, a discernible systematic trade-off emerges between the synoptic and MISO/MJO contributions within
the model. This trade-off likely plays a role in explaining the model's comparatively lower prediction skill.

Using multiple correlation analysis, we further investigate the impact of major global predictors, including ENSO, AMO, IOD, and PDO, on sub-seasonal components. Our analysis reveals significant patterns of association between these predictors and sub-seasonal bands (see Figure 9), reaffirming the findings of Saha et al. (2021). Observations show that, in the ISM region, the synoptic bands have the strongest association with these predictors. However, the 10-20 day and 20-60 day bands
in the PAC region demonstrate more pronounced connections with the predictors. As expected, the model shows a wide range of associations between sub-seasonal components and predictors. A notable deviation is observed in the PAC region, where 2-5 day band and 20-60 day bands overestimate their associations with the predictors. Analogous to the patterns observed in Figure 7a and b, the multiple correlation distinctly separates good and poor ensemble members for PACR rainfall (Figure S2). However, for the ISMR, this distinction is not very prominent. Nonetheless, this analysis underscores the significance of major
global predictors influencing sub-seasonal components, thereby elucidating the intricate dynamics that exert an influence on seasonal prediction skills.

Therefore, a significant association of sub-seasonal rainfall variance with the seasonal mean rainfall and four major global predictors is found in observations and, to a varying degree, in model simulations. Conversely, the model's shortcomings in capturing interactions between fast (i.e. sub-seasonal) and slowly evolving components (e.g. ENSO) influence the seasonal
mean and its variability. Consequently, the feasibility of estimating the PPL based on the 'perfect model' framework (or ANOVA) appears inadequate. We recall that the 'perfect model' framework assumes that the ensemble spread is due to an error in the initial conditions, and error growth due to imperfect physics or numerical scheme is not considered. It is crucial to acknowledge that real-world models are imperfect, and the realization of a truly perfect model is highly unlikely. Hence, the question arises: How can we obtain a reliable estimate of PPL that circumvents paradoxical behaviour?



**Figure 8.** Mean sub-seasonal rainfall variances simulated by the model and observations (1997-2021). Variances in the 2-5, 10-20, and 20-60 day bands with respect to the total variance (in %) over a) the ISM domain and b) the Pacific domain. c) and d) are the same as a) and b), respectively, but show their actual values..





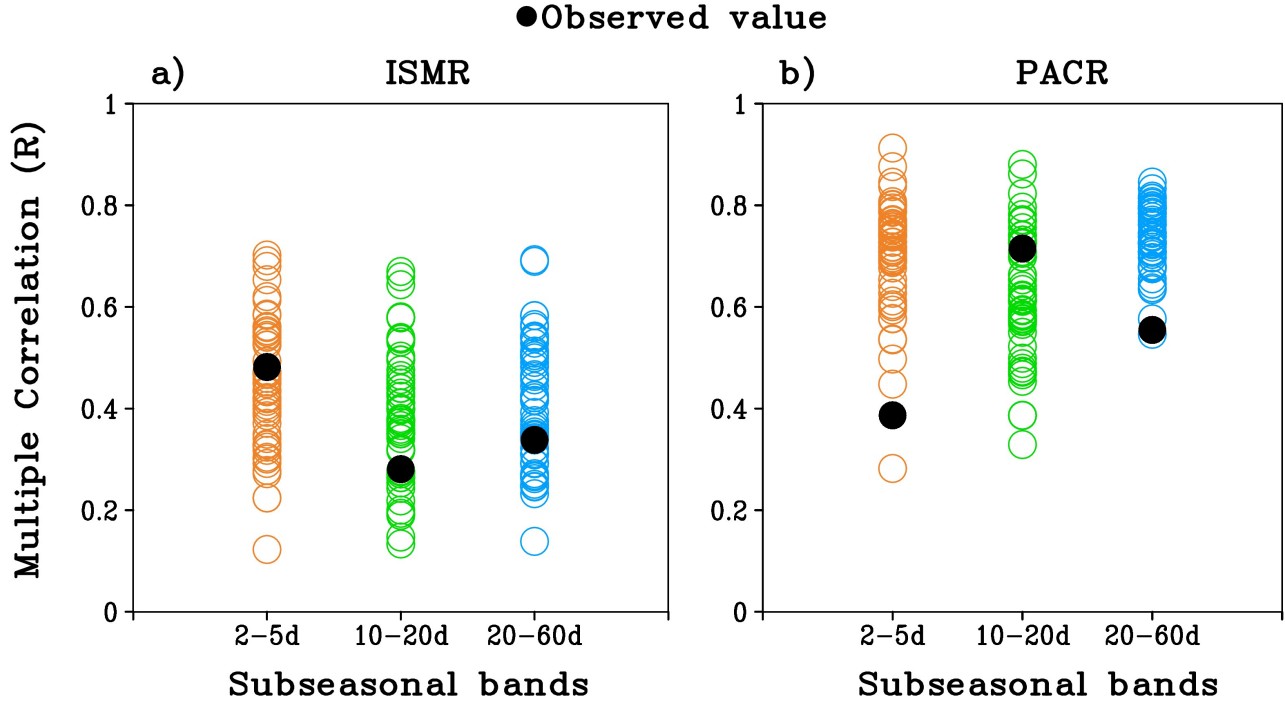

**Figure 9.** Multiple correlation analysis of sub-seasonal components (three bands) regressed with four global predictors (Niño3.4, IOD, PDO, AMO) for a) ISMR and b) PACR rainfall. Open circles represent individual ensemble members of the model, and black solid circles represent those in the observations.

### 3.4 Maximum achievable skill by model

Here, we explore the possibility of having a model's PPL that is free from paradox. Forecast errors arise from various sources, including imperfect representations of physical processes, limitations in numerical methods, and inaccuracies in the initial conditions (ICs). Among these, uncertainties in the ICs can lead to substantial variations in forecast skill. Consequently, some ICs may result in more accurate predictions than others.

Thus, even when using the same forecasting model, different sets of ICs can produce ensemble forecasts with varying skills. The concept of the PPL aims to estimate the upper bound of prediction skill achievable by a forecast system, which is not a guaranteed outcome but the maximum possible skill under current model capabilities. A large ensemble of ICs can be used to generate a distribution of re-forecast skills for a given model. Since perfect initial conditions are practically unattainable, the best achievable forecast skill estimated from a sufficiently large ensemble will always remain below the idealized limit (i.e., correlation = 1.0).





In this context, forecast skill is quantified as the correlation between observed and ensemble-mean ISMR or PACR. Accordingly, we define the maximum correlation skill obtained from a comprehensive set of ICs as the actual PPL of the forecast system. The following points clarify the rationale behind this interpretation and ensure the concept remains logically consistent:

1. Predictability refers to the intrinsic potential for a system to be predicted, while the PPL represents the maximum achievable prediction skill by a specific forecast system.

2. The PPL is inherently model-dependent, varying with model physics, dynamics, and the characteristics of the initial conditions. It is, therefore, not a universal constant but specific to each forecast system.

3. Within a fixed model and data assimilation framework, variations in prediction skill arise solely from the different realizations of initial conditions derived from the same assimilation system.

It is thus essential to establish a robust method for determining the maximum achievable skill of a forecasting system at its current stage of development. This involves leveraging a wide range of initial conditions derived from advanced data assimilation systems. Among these, some ensemble combinations may exhibit notably high skill. Hence, the maximum skill attainable by the model, computed from the ensemble subsets can be considered its PPL. This approach also helps avoid paradoxical outcomes in predictability assessments. To achieve this, the maximum prediction skill is evaluated through correlations between observations and ensemble means, considering all possible combinations of ensemble members, i.e., $^{n}C_{r}$, where n is the total number of ensemble members, and r is the number used in each subset.

However, the use of a large number of ensembles can impose significant computational demands. For instance, with a set of 52 ensembles, there are approximately $4.9 \times 10^{14}$ sub-sets of 26 ensemble members (without repetition of any ensemble member in a subset). An alternative approach is to generate a limited number of random sub-sets and determine the maximum skill. To illustrate this feasibility, we have selected the best 40 out of 52 ensembles for 41 years (1981-2021) and computed correlations between ensemble-averaged ISMR rainfall and observations for all possible combinations (up to a maximum of $1.378 \times 10^{11}$ subsets). Figure 10a depicts the maximum, minimum, and mean correlations across various combinations (ranging from subsets of 2 to 40 ensemble members). Notably, it becomes apparent that a combination of 5-10 ensemble members yields a maximum correlation skill of approximately 0.76.

Now, we examine the possibility of using a reduced number of random combinations (up to a maximum of $10^{6}$) while ensuring that ensemble members do not repeat within a sub-set or combination. Remarkably, similar patterns emerge in terms of maximum, minimum, and mean correlations when using randomly selected subsets from the full spectrum of possible combinations (Figure 10a). It is noteworthy that the mean, minimum, and maximum correlations tend to converge as the number of combinations increases. This phenomenon is associated with the increase in the number of combinations from 2 to 20 and subsequent decrease, ultimately culminating in only one combination at 40 ensemble members. Nonetheless, the adoption of random combinations proves to be a cost-effective approach that does not compromise the quality of results.

Sampling limitations often hinder accurate estimation of population-level statistics, as available model or observational data may not fully represent the true statistical characteristics of the entire population. To address this, bootstrapping, a resampling





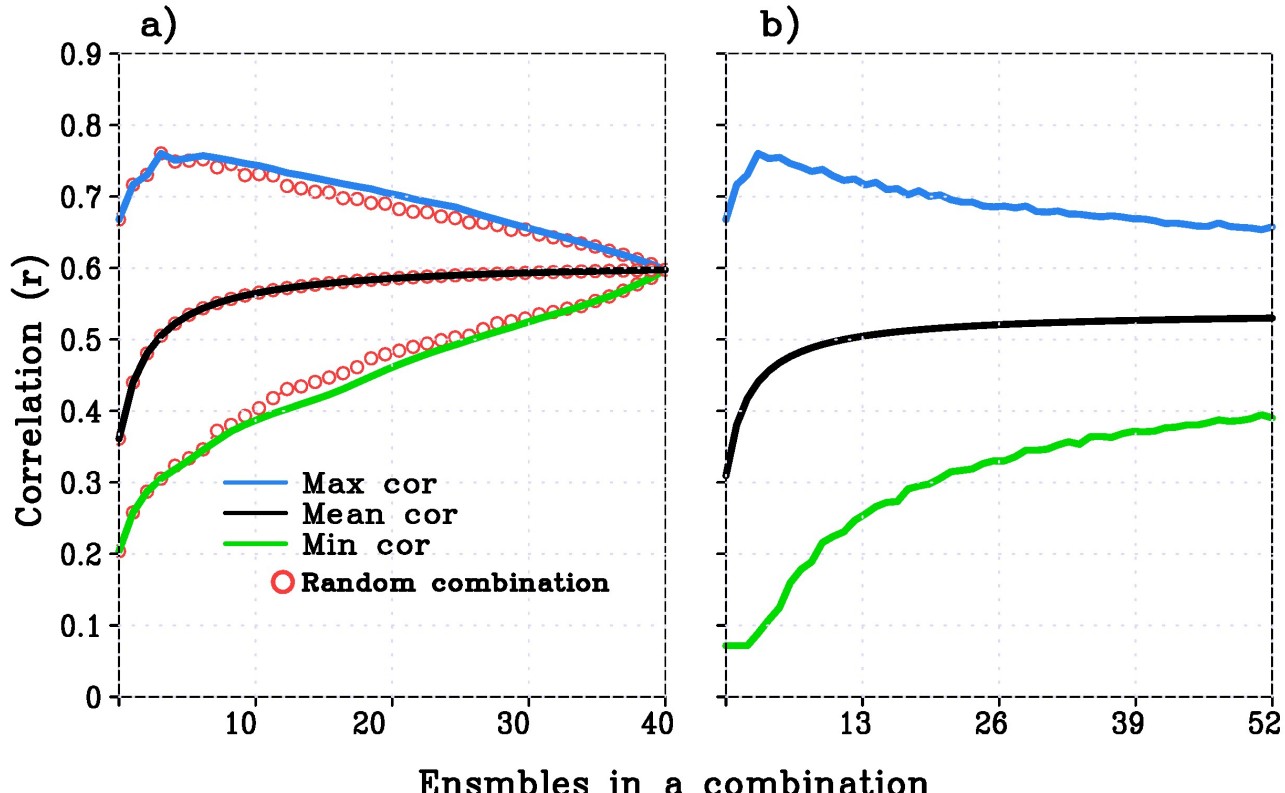

**Figure 10.** a) The maximum (blue solid line), mean (black solid line), and minimum (green solid line) correlation skill using all combinations of n ensemble-averaged ISMR rainfall (i.e., $^{40}C_n$, where n varies from 1 to 40). Similarly, the red open circles show the correlation skill using 1 million random combinations of n ensemble-averaged ISMR rainfall. b) The maximum (blue solid line), mean (black solid line), and minimum (green solid line) correlation skill using the Bootstrapping method with 10 million random combinations of n member ensemble-averaged (n varies from 1 to 52).

technique, offers an effective approach to derive more accurate population statistics from such sub-sampled data. In this study,

390    we utilize 52 ensemble members over 41 years of re-forecasts, which may be insufficient to capture the model's full predictive potential. Since our goal is to estimate the maximum forecast skill, bootstrapping serves as a practical alternative to exhaustive combination methods.

     Unlike fixed combinations where ensemble members are non-repeating, bootstrapping allows repetition, enabling the creation of millions of subsets of ensemble members. We employed 10 million such subsets, varying in size from 2 to 52 members,

395    and calculated the maximum, minimum, and mean correlation skill (Figure 10b). Interestingly, the highest skill observed is approximately 0.76 for subsets of 5 members and around 0.66 for the full set of 52 members.





The same methodology can be extended to the grid level, enabling the estimation of the global pattern of PPL. Although the maximum skill for ISMR rainfall is achieved with approximately a 5-ensemble member subset (correlation coefficient $\sim 0.76$), we have chosen to use a 10-ensemble member subset to ensure a sufficient number of ensembles for PPL estimation at each grid point. Maximum correlation skill values are recorded at every grid point, derived from $10^5$ combinations or subsets of 10 out of the 52 ensemble members using the bootstrap method. Consistent with the PPL estimates obtained through the ANOVA methods, the same three variables (rainfall, MSLP, SST/LST) have been employed in this analysis (Figure 11). It is clear that the highest attainable skill for rainfall is situated over the equatorial Pacific region, reaching a maximum of approximately 0.9. This is consistent with previous studies (e.g. Shukla, 1998) that have consistently demonstrated maximum prediction skill in this area compared to other regions of the globe. Regarding MSLP, skill appears to be more pronounced in the South Asian monsoon and eastern Pacific regions. A substantial portion of Europe, the Middle East, and northeastern Asia exhibits notable skill in 2-meter air temperature, with correlation coefficients ranging from approximately 0.8 to 0.9. It's worth noting that this method does not suffer from the paradox problem, as the maximum correlation value obtained by this technique is higher than the correlation skill, and this holds true for all grid points. Areas showing poor maximum skill or PPL can be targeted for further improvements, given that most ensemble members or their combinations demonstrate bad performance.

## 4    Summary and discussions

In this study, we employ a data set comprising 52 ensemble member re-forecasts spanning a 41-year period from 1981 to 2021, as simulated by the IITM CFS (Saha et al., 2019). The primary objective of this study is to elucidate the underlying factors contributing to the signal-to-noise paradox in seasonal climate prediction within the tropical and subtropical regions. We observe that many regions exhibit a signal-to-noise paradox when estimated by subtracting correlation skill from potential predictability estimated by the ANOVA method (Figure 1). This region also corresponds to areas where the ratio of predictable component (RPC) exceeds one. RPC is an alternative measure of predictability, with values surpassing one being deemed regions exhibiting paradoxical behaviour. Several prior studies have highlighted the presence of paradoxes in PPL estimates within the 'perfect model' framework (i.e. ANOVA ). However, the underlying causes remain a subject of ongoing debate within the scientific community. As paradoxical behaviour is rare in weather time scale but prevalent on seasonal to decadal and beyond time scale, assessment of the underlying hypothesis for the method of estimating PPL is warranted. Therefore, to delve deeper into the intricacies of this paradoxical behaviour, we examine rainfall patterns in two distinct tropical regions: the Indian summer monsoon region (ISMR), exhibiting paradoxical behaviour over half of the region, and the central Pacific region (PACR), without any paradox.

While correlation skill in the time series of seasonal ISMR and PACR rainfall decreases from relatively good to poor ensemble members (arranged with respect to observations), the PPL exhibits minimal variation (Figure 3). This suggests that PPL is not closely associated with the actual skill of a model, contradicting the fundamental purpose of estimating it. In the ANOVA method, the separation of a forecast into 'signal' and 'noise' components is based on the assumption that initial error gives rise to the 'noise', while the model is considered to be perfect. Consequently, there should be no temporal relationship





**Figure 11.** Maximum Correlation at each grid point using 100 thousand Bootstrapped re-sampling of 10 ensemble average from 52 ensembles and 41 years (1981-2021) simulation by IITM-CFS. Maximum correlation in JJAS-averaged a) Rainfall, b) Mean Sea Level Pressure, and c) surface temperature (SST over Ocean and 2m temperature over land).





between 'noise' and 'signal' components. Conversely, it is observed that a significant and robust correlation exists between these components. Furthermore, regions exhibiting paradoxical behaviour do not necessarily coincide with areas of significant correlation between the 'signal' and 'noise' (Figures 2).

Hence, the central issue underlying this problem is why the ANOVA method falls short in estimating climate predictability? What leads to the apparent inadequacy in separating a forecast into 'signal' and 'noise' components? Some of the answers

can be found in the nature of the problem. Weather forecasting pertains to the predictability of the first kind, where the accuracy of initial conditions plays a primary role, and slowly evolving boundary conditions play a secondary role. Conversely, seasonal/decadal forecasting is categorized as predictability of the second kind, where slowly evolving boundary conditions play a major role. Moreover, the physical processes which affect the sub-seasonal variability at various frequencies (e.g. lows and depressions, MJOs, etc) can also affect the seasonal mean. As the real world's models are not perfect, these deficiencies

can also add to the variability or ensemble spread in addition to that due to errors in the initial conditions.

Therefore, understanding the co-variability between sub-seasonal components and seasonal means within the context of predictability and prediction is of paramount importance. While prediction skill demonstrates systematic variations with the covariance between sub-seasonal components and seasonal mean rainfall, the PPL remains quite invariant (Figure 3). It becomes evident that the model exhibits a broad and often skewed range of co-variability between sub-seasonal variance and

seasonal mean in comparison to observations, with sub-seasonal components strongly linked to global predictors (see Figures 7, 9,). Additionally, ensemble spread (or noise component) is found to vary systematically with this co-variance and prediction skill (Figure 3, 7). Consequently, a robust methodology for estimating PPL must consider not only the errors in initial conditions but also those arising from model physics, dynamics, and numerical methods employed. These findings underscore the critical significance of accurately simulating sub-seasonal components and their co-variability in seasonal prediction, shedding

light on factors influencing prediction skill and identifying potential areas for model enhancement.

Building upon the work of Saha et al. (2019), we employ a simple method to estimate the maximum prediction skill by a model. This method serves to determine the upper bound on seasonal climate prediction skill attainable by a model. In consideration of the constraint posed by a finite sample size (i.e., 52 ensembles for 41 years), and its potential impact on the accuracy of potential predictability estimates, we employ the Bootstrapping method. This statistical technique enables

us to infer the maximum prediction skill of the entire ensemble population by re-sampling from the available ensembles, thereby mitigating the influence of sample size limitations on our estimates. Since estimates of PPL are model-dependent and subject to change with improvements in the model and initial conditions (e.g. Kumar et al., 2014), we propose that the maximum prediction skill obtained through our method represents the model's PPL at its current state of development and initial conditions. This approach effectively avoids paradoxical situations.

*Code and data availability.* The code and scripts used in the preparation of this manuscript are publicly available in Yashas (2025) https://doi.org/10.5281/zenodo.15369106. The observational data sets used in the study are: 2m air temperature data over land from the Climatic Research Unit (CRU TS3.1) available in Harris et al. (2014). SST data is taken from EN4 reanalysis, which is available in Good et al. (2013).



MSLP data is obtained from ERA5 reanalysis, available freely in Hersbach et al. (2020). Daily rainfall data from the Global Precipitation Climatology Project (GPCP Version 1.3) available in Huffman et al. (2001) for 1997-2021 and monthly rainfall data from GPCP version
2.3 for 1979-2021 available in Adler et al. (2020) . CFS re-forecast of monthly SST, MSLP, surface temperature and daily precipitation for June-to-September (1981-2021) with 52 ensemble members are publicly available in Saha (2024) https://doi.org/10.5281/zenodo.13166897. Freely available software GrADS is used for plot and data analysis (http://cola.gmu.edu/grads/).

*Author contributions.*  YS and SKS conceptualized the idea and carried out model simulation and data analysis, and wrote the manuscript. SP, MK and UV contributed to the discussion, plotting Figures and manuscript writing.

*Competing interests.*  There are no competing interests.

*Acknowledgements.*  We thank MoES, the Government of India and the Director IITM for all the support in carrying out this work.



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
