# Peer review of "Why does the signal-to-noise paradox exist in seasonal climate predictability?"

_EGUsphere, 2025_

## Referee Comment (RC1)

Dear Dr. Neale

I have carefully reviewed their response and am deeply disappointed. Not only did their reply fail to address the key points, but more importantly, the authors appear to be responding to non-existent arguments (as noted in my comments below). I believe this study is methodologically flawed and conceptually confusing. Therefore, I am convinced that this work should not be published.

Below is my response to their replies. The red text represents their original comments, the bold black text is my previous feedback, and the blue text contains my current responses.

We thank you for taking your time and giving your comments, which are useful for improving the manuscript. Here are point wise clarifications.

**1. The article only explains how signal and noise variance are defined and calculated. Since variance itself is not the actual component, it is unclear how the signal and noise are extracted from the data. The concept and defintion are totally different between the variance and the variable itself.**

Reply: There are numerous paper on how signal and noise components are extracted from model data and some of them are cited here (e.g. Kang and Shukla, 2006; Scaife et al., 2014; Saha et al., 2016a; Scaife and Smith, 2018; Weisheimer et al., 2018 and many more). While inter-ensemble spread is considered as noise/internal component, the ensemble mean is the signal/external component (equation 1 and 2 respectively in our manuscript). How signal and noise are extracted from data is clearly mentioned in lines 104-108 of the manuscript, section 2.3.1.

The articles cited by the authors only discuss signal variance and noise variance. It is problematic to treat the ensemble mean directly as the signal/external component. As a measure of variability, the signal should not be constrained by sign—how does one interpret a "positive signal" versus a "negative signal"? Therefore, it is more appropriate to use the square of the ensemble mean to represent the signal.

In the author's statements in lines 104–109 as below, I could not find a clear definition of either the signal or the noise.

respectively. Here, predictable and unpredictable components are termed external/signal and internal/noise components, respectively. The ratio of external to internal variance is known as the signal-to-noise ratio (SNR). If x is the precipitation field of the model, i is the year of the model integration (total year 'N'), and j is the number of ensemble simulations (total ensemble n = 52), then internal variance following Rowell et al. (1995), can be expressed as

**2. The article tries to discuss and analyze the paradox, but the purpose of using Nino3.4 to predict precipitation remains unclear. What is the intention behind comparing it with dynamic models? Is it to demonstrate whether the actual or potential forecast skill of dynamic models is higher or lower, reasonable or unreasonable? The objective is not clearly stated. Moreover, can using Nino3.4 to predict precipitation effectively achieve these goals? Would the forecast skill be reliable? Was the forecast skill mentioned in the article derived from training or test data? Similarly, were other modes affecting precipitation in the Indian region, such as IOD, considered?**

Reply: The idea is to asses prediction skill of not only predictants (i.e. ISMR, PACR), but also the fidelity in simulating global predictors (e.g. ENSO) and their teleconnections. Figure 9 shows multiple correlations involving major global predictors (Niño3.4, IOD, PDO, AMO) and sub-seasonal components.

Your response does not address my question. Such a simple linear regression approach is unreliable and insufficient to explain any core issues discussed in this paper.

**3. Rowell (1995) never defined signal variance and noise variance using ANOVA. While they did mention ANOVA, it was only used for statistical testing. The authors should revisit Rowell (1995) to better understand the content. ANOVA has exactly defintion in statistics, which should be followed to avoide unnecessary confusion.**

Reply: Please look into page no 699 of Rowell et al. (1995).
https://rmets.onlinelibrary.wiley.com/doi/epdf/10.1002/qj.49712152311

,which mention "The approach we use to estimate the components of variance closely follows an 'analysis of variance' methodology ..."

I could not find the answers provided by the authors in Rowell (page 699) as below. It should be noted that ANOVA has a rigorous statistical definition. The authors, however, only performed variance partitioning, not ANOVA.

The next stage of research will be to explore the physical mechanisms which link the SST patterns to seasonal rainfall variability. Circulation changes over north Africa will be examined in a later publication, and some global-scale circulation patterns associated with Sahelian rainfall anomalies are presented by Ward *et al.* (1994).

Given that SST patterns are often predictable at least a few months in advance, this offers hope for the production of skilful forecasts of seasonal JAS rainfall anomalies averaged over the Sahel, Soudan and Guinea Coast. Indeed, such forecasts have now been issued by the UK Meteorological Office for the Sahel region since 1986, and for the Soudan and Guinea Coast regions since 1992, on an experimental basis (see Ward *et al.* (1993) for details). In order that such forecasts achieve maximum utility, further research is required on the variations of rainfall–SST relationships *within* the large regions used here and *within* the July to September season.

ACKNOWLEDGEMENTS

**4. I do not understand the meaning of the statement: "The use of the orthogonality assumption is a methodological simplification to partition variance across time scales; it does not imply the absence of physical co-variability." Do physical and mathematical co-variability have different interpretations? In my opinion, if two quantities are physically related, they cannot be assumed to be orthogonal in mathematics. Additionally, I do not comprehend the authors' claim that "sub-seasonal components are the building blocks of the seasonal mean." Following this logic, all time scales would be sources of error, since hourly components are the building blocks of the daily mean, and daily components are the building blocks of the weekly mean, and so on.**

Reply: The argument why we are using assumption of orthogonality and not the actual one, lies on the fact that it is challenging (if not impossible) in a non-linear system to separate individual components.

It seems no basis to argue the "challenging to separate" as a justification for such an assumption. This is the most critical weakness of the study: on the one hand, it attempts to examine the effect of A on B using linear statistical analysis, while on the other hand, it assumes that A and B are orthogonal, implying that their covariance (or correlation coefficient) is zero.

Sub-seasonal components of the monsoon particularly have clear preferred band. Some of the band are more vigorous in terms of their spatial scale, strength than the others. In terms of their contribution to the mean and variability/predictability also varies. While MISOs have very large spatial structure and strong sub-seasonal variability, their contribution to year-to-year monsoon rainfall variability is minimum (weak negative correlation). So, clearly, we are not talking here about hourly/daily events but some known and prominent sub-seasonal variability/bands, which shape the seasonal monsoon rainfall of a year. Here are literatures, cited in support of our arguments (Saha et al., 2019; Borah et al., 2020). Some important papers in the similar lines but not cited here are.

I am drawing this inference based on the authors' own argument. You may choose to ignore or omit other scales of the atmospheric process, but I cannot overlook them. Isn't that ?

**5. So I have to feel sorry to decline this work again. The topic is interesting that is the reason why I agreed with reviewing it. Unfortunately I do not learn more from this work. To my understanding, the paradox should be from the "defintion" of potential predictability. The ratio of signal to noise may not well represent the potential predictability. If authors wish to work this problem, I suggest them to seek other measures to quantify the potential predictabilty.**

Reply: We wish, if you could have read the full manuscript. The main content of the manuscript is the following:

i) Perfect model framework is used to estimate potential predictability of seasonal anomaly, which often shows paradoxical behaviour. 'Analysis of variance' framework is used for calculating 'signal' and 'noise' components using 52-ensemble member re-forecasts.

ii) Here we argue that 'perfect model framework' is not adequate, as the error growth is not from only initial condition errors but also from other sources, like physics, numerical scheme etc. We demonstrated that sub-seasonal component, which is part of the physics, adds error (biased contribution) in the seasonal forecast anomaly (i.e. Figure 7). However, 'perfect model framework' assumes, ensemble spread solely attributed to initial condition error. Consequently, true limit of predictability is not known. So, here our argument matches with your point of view that the method of

estimating PPL based on perfect model framework is inadequate. We have already mentioned it in lines 337-344, in the last para of section 3.3

The authors appear to lack a clear understanding of the PPL issue. PPL is fundamentally a product of the "perfect model" framework. Once model errors are taken into account, it ceases to be a PPL problem. Therefore, the very premise of this study is conceptually inconsistent.

iii) Finally we propose a method for estimating PPL, which is free from paradox (section 3.4). Therefore, we believe the rationale provided for rejection does not fully capture the merits of the manuscript

---

## Community Comment (CC3)

With my own interest in Indian monsoon prediction and predictability, I read the manuscript with interest. The manuscript addresses an important and timely issue concerning the potential predictability limit (PPL) of the seasonal monsoon and the inherent challenges associated with the 'perfect model' framework. The question of how far seasonal climate, particularly Indian summer monsoon rainfall, can be predicted remains a subject of active debate. The study is highly relevant to the climate dynamics and prediction community, with potential implications for both advancing scientific understanding and improving operational forecasting. Overall, the manuscript is well-structured, presents new insights, and uses a comprehensive set of model simulations. I am happy to recommend the manuscript for publication after minor revision.

**Strength:**
1) In the perfect model framework, signal and noise component of a parameter is estimated, assuming that ensemble spread is due to initial error, which is more appropriate for short-range forecast (e.g. weather), but may not hold true for long-range forecast (i.e. seasonal), where noise/error introduced by slowly varying boundary conditions are also important. Therefore, estimates of PPL for seasonal climate in this framework may not represent the true limit, which is all about paradox here.

2) Figure 7 shows an important aspect: how the internal variability could contribute to the prediction skill/predictability of ISMR. However noise component is fully attributed to initial error in 'perfect model' assumption.

4) Although signal and noise are estimated under the assumption of orthogonality, it is clear that this assumption does not always hold.

3) Interestingly, proposed method of estimating PPL and ANOVA based PPL are similar in their maximum predictability of rainfall over tropical Pacific region and more importantly free from paradox.

**Weakness:**
1) PPLs are model dependent, improvements in model likely to increase the limit. Longer observations may be required to estimate actual PPL.

**Suggestions:**

1) For readers from other domain, some basic discussions, like what is the basic premise/hypothesis of the 'perfect model' framework while using ensemble forecast is required.
2) The estimate of PPL by using a seasonal prediction model from a large ensemble of hindcasts by choosing the ensemble mean of 'best' initial conditions may be acceptable. However, the manuscript does not provide a discussion on how to realise the PPL in operational framework. It is possible that growth of 'initial error' may never allow the model to achieve the PPL. Even if we knew what are the 'best initial conditions' in the ensemble, it would lead to overfitting and unreliable forecast. A discussion on how the PPL could be achieved either by tradition methods or by a deep learning/AI model trained on the large ensemble of hindcast experiments would significantly enhance the quality of the manuscript.

---

## Community Comment (CC5)

**Strength:**

**1) In the perfect model framework, signal and noise component of a parameter is estimated, assuming that ensemble spread is due to initial error, which is more appropriate for short-range forecast (e.g. weather), but may not hold true for long-range forecast (i.e. seasonal), where noise/error introduced by slowly varying boundary conditions are also important. Therefore, estimates of PPL for seasonal climate in this framework may not represent the true limit, which is all about paradox here.**

**2) Figure 7 shows an important aspect: how the internal variability could contribute to the prediction skill/predictability of ISMR. However noise component is fully attributed to initial error in 'perfect model' assumption.**

**4) Although signal and noise are estimated under the assumption of orthogonality, it is clear that this assumption does not always hold.**

**3) Interestingly, proposed method of estimating PPL and ANOVA based PPL are similar in their maximum predictability of rainfall over tropical Pacific region and more importantly free from paradox.**

Dear Prof Goswami,

We thank you for your constructive comments on our manuscript. We value your recognition of the study's contributions. And appreciate mentioning the highlights of our work. Here are reply point wise, which we are going to incorporate in the next version of the manuscript to address all the concerns and suggestions. We thank you for highlighting these key aspects of our work.

**Weakness:**

**PPLs are model dependent, improvements in model likely to increase the limit. Longer observations may be required to estimate actual PPL.**

We agree with this statement. We have acknowledged these points in the revised manuscript.

**Suggestions:**

**For readers from other domain, some basic discussions, like what is the basic premise/hypothesis of the 'perfect model' framework while using ensemble forecast is required.**

Thank you for this suggestion. We have now provided the context about perfect model framework in the introduction.

**The estimate of PPL by using a seasonal prediction model from a large ensemble of hindcasts by choosing the ensemble mean of 'best' initial conditions may be acceptable. However, the manuscript does not provide a discussion on how to realise the PPL in operational framework. It is possible that growth of 'initial error' may never allow the model to achieve the PPL. Even if we knew what are the 'best initial conditions' in the**

**ensemble, it would lead to overfitting and unreliable forecast. A discussion on how the PPL could be achieved either by tradition methods or by a deep learning/AI model trained on the large ensemble of hindcast experiments would significantly enhance the quality of the manuscript.**

We agree that this method is a diagnostic/post-hoc sensitivity tool rather than a ready-to-implement operational algorithm. In real-time forecasting, growth of initial condition errors, along with model physics errors make it difficult to achieve PPL. Selecting "best" members in real time would lead to overfitting. We will include a discussion on how PPL could be achieved by other methods like AI/deep learning technique.

We believe these changes will address your concerns and improve the manuscript's clarity. We will submit a revised version incorporating these revisions shortly. Please let us know if you have any further suggestions.

Thanking you,

Yashas Shivamurthy

---

## Community Comment (CC6)

**I think this study has value, for example in pointing out the lack of subseasonal variance in the model, and trying to dispel the continuing myth about PPL being an upper limit.**

Dear Prof Adam Scaife,

Thank you very much for your constructive comments on our manuscript. We greatly appreciate the time you have taken to review it. We value your recognition of the study's contribution. Below, we address major and minor points in detail, outlining our responses and proposed revisions to the manuscript.

➢ **Major points:**

**L146-147, L188, L281, Fig.6: I don't think we can make the general statement that seasonal precipitation is only modulated by sub-seasonal components. This is one of the points of the reviewer and I think it has some weight. Similarly, it is not clear that you can just dismiss the first term on the right of eqn12. On this, I think reviewer 1 also has a point. In fact there is evidence this term could be large, for example for the NAO (see Keeley et al GRL 2009 for an illustration of where it is the interannual variability per se and not the shorter timescales that dominates). I therefore don't think L188 is justified in general.**

We agree that our statement regarding the modulation of seasonal precipitation by sub-seasonal components may have been overly general and not applicable universally. Our intention was to emphasize this mechanism in the context of tropical and subtropical precipitation (e.g., during the boreal summer monsoon), where sub-seasonal events like synoptic systems (e.g. Yoon and Chen (2005)) and intra-seasonal oscillations (e.g. Goswami et al., 2006; Webster eet al. 1998) play a dominant role in building the seasonal mean.

Unlike variables such as temperature or surface pressure, which vary smoothly in time, rainfall is inherently discrete and typically occurs in pulses (rain or no-rain) concentrated within preferred time bands (i.e., sub-seasonal bands). Furthermore, the amplitude of these events is often much larger than that of the annual cycle. As a result, variations in sub-seasonal rainfall can significantly modify the annual cycle or seasonal anomaly.

To illustrate this, here we use daily 1°×1° IMD rainfall over a grid point in central India (20°N, 80°E), a homogeneous monsoon region. The amplitude of the climatological mean annual cycle (1901–2018) is about 20 mm/day (upper panel). The daily rainfall and corresponding annual cycle for a particular year (here, 2002; lower panel) show strong temporal fluctuations with large amplitudes (blue). The smooth annual cycle is reconstructed using the mean and the first three harmonics. The difference between the climatological mean annual cycle (black curve) and the annual cycle for 2002 represents the seasonal summer monsoon rainfall anomaly (a deficit monsoon year).

[Figure]

**Figure R1:** *Climatological mean daily rainfall (black solid bar) and smooth annual cycle (black line) over an area in the homogeneous central India region are shown in the upper panel. Lower panel shows rainfall of a particular year (here 2002) with smooth annual cycle (blue line), smooth annual cycle after removing two days of rainfall events (red line) and climatological mean smooth annual cycle (black line).*

To demonstrate how rainfall event of just one or two days could influence the annual cycle and seasonal anomaly, two-day rainfall event (>80 mm/day) are removed, assuming it arises from sub-seasonal variability. The reconstructed annual cycle, without these two days rainfall becomes visibly weaker (red curve). The resulting change in seasonal mean rainfall amounts to about **59%** of the interannual standard deviation. We also note that only two 1-day rainfall events are removed here; if a complete event is removed, as happens, the impact on the seasonal anomaly would be substantially larger. This highlights why sub-seasonal components are often termed the "building blocks" of the monsoon.

Because rainfall is a discrete phenomenon, it does not possess true, physically persistent modes. Thus, global predictors influence monsoon rainfall primarily by modulating the sub-seasonal components either through their strength, their duration, or both. The "persistent modes" that

emerge from various data-analysis techniques are projections or statistical composites of these sub-seasonal rainfall components. We have now modified the manuscript with above discussion and added above figures in the supplementary section.

Regarding the first term on the right-hand side of Eq. (12), we acknowledge that the earlier description may have caused confusion. The seasonal anomaly is treated as an external term, under the assumption that it is entirely generated by the predictors, which act as the drivers of the seasonal anomaly. To address this issue, we have revised the manuscript. Now the manuscript is modified as

The time series of daily rainfall of a year (area average or a single point) can be represented by the following equation

$$x_T = x_c + x_a + \sum_f x_f \tag{7}$$

where, $x_T$ is the total rain, $x_c$ is the climatological mean annual cycle, $x_a$ is the anomalous annual cycle, $x_f$ represents the rest sub-seasonal components consisting of all frequencies $f$. Using harmonic analysis, the sum of the mean and the first three harmonics represents the 'smooth annual cycle' in the daily time series for a year. Here, $x_c$ is the climatological mean of the 'smooth annual cycle', and $x_a$ is the deviation of the 'smooth annual cycle' of a year from the climatological mean annual cycle. Sum of $x_a$ in a season is the exact seasonal anomaly. Therefore, after re-arrangement, the above equation can be written as

$$(x_T - x_c) = x_a + \sum_f x_f \tag{8}$$

The left-hand term represents the total daily anomaly. In terms of seasonal variance, using daily June-to-September data (122 days) equation 8 for a particular season can be written as

$$\sum_{l=1}^{122}(x_T^l - x_c^l)^2 = \sum_{l=1}^{122}(x_a^l)^2 + \sum_{f=1}^{K}\sum_{l=1}^{122}(2x_a^l.x_f^l) + \sum_{f=1}^{K}\sum_{l=1}^{122}(x_f^l)^2 \tag{9}$$

$$V = V_a + \sum_f V_{cov} + \sum_f V_f \tag{10}$$

where $l$ represents the day, $V$ is the total variance, $V_a$ is the variance of the anomalous annual cycle, $V_{cov}$ is the covariance among sub-seasonal and anomalous annual cycle, $V_f$ represents the sub-seasonal variance, $K$ is the number of sub-seasonal bands (e.g., synoptic, bi-weekly) in a season. However, due to orthogonality, the covariance term becomes negligible. In terms of seasonal anomaly, equation 10 can be written as

$$V' = V_a' + \sum_f V_f' \tag{11}$$

Where, $V'$, $V_a'$ and $V_f'$ are seasonal anomalies of the total variance, variance of anomalous annual cycle, and sub-seasonal variance respectively. Let $I'$ be the anomaly of seasonal-mean rainfall then, the covariance between the seasonal rainfall anomaly and anomaly of total variance can be written as

$$\sum_i V'I' = \sum_i V_a'I' + \sum_i \sum_f V_f'I' \tag{12}$$

The left-hand term of equation 12 represents the interannual covariance between total sub-seasonal variance and seasonal anomaly. The first term on the right-hand side represents the covariance between variance of anomalous annual cycle and the seasonal mean, while the second term represents the covariance between variance of rest sub-seasonal components and the seasonal mean. It is important to note that the first term on the right-hand side explicitly does not contain information on the building blocks of the seasonal mean and is, therefore, not used in our analysis. On the other hand, the last term is of particular interest, as it represents the interannual covariance between seasonal mean and sub-seasonal bands.

Although the anomalous annual cycle ($x_a$) and sub-seasonal components ($x_f$) are orthogonal, their seasonal variances (equation 11) are interlinked on year-to-year time scale (Figure R2), indicating role of sub-seasonal components on seasonal anomaly (Figure R2). Correlations > 0.35 (< -0.35) are significant at 95% level.

[Figure]

*Figure R2: Linking sub-seasonal components of ISMR with anomalous annual cycle (i.e. seasonal anomaly) in terms of their variances. Moving window correlation (31-years) between $V_a$ and $V_f$ (20-90 days, 10-20 days, <10 days band) and multiple correlation.*

The variance of sub-seasonal components in a season represents its energy or vigour, which also, in principle, should be linked with seasonal rainfall anomaly (i.e. last term in equation 12). A strong correlation of all India seasonal rainfall (i.e. ISMR) anomaly with variance of individual sub-seasonal components (Figure R3) support our arguments that sub-seasonal components are key to generating seasonal anomaly.

[Figure]

**L234 and abstract: I don't think anyone is saying that partitioning into signal and noise is going to eliminate paradoxical behaviour so I don't really understand this line.**

Thank you for pointing this out. We did not intend to claim that perfect partitioning into signal and noise would eliminate paradoxical behaviour entirely. Rather, our point was to highlight that even with accurate separation (as assumed in the perfect model framework), paradoxes persist due to other factors like model imperfections. We have rephrased line 234 and the relevant abstract sentence for clarity.

**L319: In this case the total variance is too strong. In our original papers we were careful to say that the paradox only really arises if the total variance is close to the observed variance – otherwise it could simply be a case of overdispersion – is that possibly the case here?**

We appreciate this insight and agree that the paradox is most pronounced when total variance matches observations, whereas overdispersion (excessive total variance in the model) could explain apparent paradoxes in some cases. In our analysis, the total variance in the model for rainfall over the study regions (South Asia Monsoon region) is indeed higher than observed (as noted in Section 3.2 and Fig. 5b, 8c, 8d, which may indicate overdispersion rather than a "pure" low signal-to-noise issue. We have expanded the discussion at Line 319 to explicitly report on total variance comparisons with observations, referencing relevant papers (e.g., Scaife et al., 2014; Scaife and Smith, 2018) to distinguish between these scenarios.

**Finally, I'm afraid I do not agree that the proposed method of selecting the ensemble members that yield the highest correlation with the observations is a viable algorithm for determining the upper limit of predictability. This can be illustrated if we consider a system with no skill but some random noise. Some combinations of ensemble members (half of them in fact) will then exhibit apparent skill even though none is present. The problem is particularly acute for small ensembles as you find in Fig.10 so I think this section should be removed.**

We thank you for your views. Here our objective is to find out the range of actual skill achievable by this particular model. With limited number of ensemble members, we may be able to find out the population statistics using random choice of ensemble members. The distribution of actual skill with possible maximum and minimum defines the model's ability to predict seasonal monsoon rainfall. Improvements of model is likely to shift the whole distribution towards higher correlation side (e.g. Figure 4b in Saha et al., 2019).

We agree that with ensemble members having no skill (half of the members negative correlation and half with positive correlation), could show higher predictability. However, this is not the case here. We have shown minimum, mean, and maximum possible skill and all of

them are positive (Figure 10). We do not propose this as a formal, universally applicable algorithm for redefining the PPL, nor as an operational technique. Rather, it is an experimental diagnostic to illustrate the maximum possible skill that can be obtained. We believe, it is an important diagnostic, which helps to understand predicative capability of a model and would like to retain in the manuscript with some modification/caveats.

We have now revised the manuscript to reflect the same. In the abstract, we have modified the sentence "In this context, we propose a novel method to estimate the PPL of seasonal climate, which can be free from paradoxical situation" to "In this context, we present a simple diagnostic approach to estimate the maximum achievable seasonal prediction skill, which may be interpreted as the PPL". Similarly in result and discussion section we have modified keeping in mind that it is just a diagnostic and not the true PPL.

> **Minor points:**

**L7: Regarding the orthogonality of noise and signal, there is a relevant recent paper by Brocker et al in QJRMS 2023 which makes a similar point. It certainly should be referenced and this may make it easier to justify the point about signal and noise not being orthogonal in time. See: https://rmets.onlinelibrary.wiley.com/doi/10.1002/qj.4440**

Thank you for suggesting this. We have included this citation.

**L30: I think that although it is not perfect, monsoon prediction skill is now well established so is this statement a little negative?**

We agree. We have modified the statement.

**L33-35: As we both know, the PPL is not an upper limit of predictability (or even the prediction skill of the model) so could rephrase to something like "…it is commonly assumed that the PPL is an upper limit on predictability…"**

Thanks for the suggestion. We have made the changes.

**L37: "….variance in models is too weak to explain the level of prediction skill."**

We have revised this statement.

**L48: "…can arise…" rather than "…arises…"**

Will Change to: "...can arise..."

**L57: skills**

Will be Corrected to "skills".

**L114: should it be "…is often an overestimate of external…."**

We will rephrase to: "...the variance of the ensemble mean is often an overestimate of external..."

**L224: I think we need to be careful about saying internal variability = noise as ENSO, the QBO etc are all internal oscillation but are still predictable on these timescales. Suggest you use "internal unpredictable variability (noise)…" or similar**

Thank you very much for pointing out this. We have now used signal and noise terms only and avoided using 'external' and 'internal' terms.

**L253-254: I don think the paradox arises from splitting into signal and noise because the other measure we use is the ratio of Rmo/Rmm in Scaife and Smith 2018 which also exceeds 1, again showing the paradox but with no separation into signal and noise.**

We agree, the paradox persists even without explicit signal-noise separation. We have tested the possibility from the point of view how signal and noise are estimated. As the current method considers only role of initial error and no other sources of error, the estimate of signal and noise is not accurate. As a result, signal and noise are not orthogonal. From this perspective also, it suggests perfect model framework is not adequate for estimating PPL. We have now revised the manuscript and included discussion related to finding using Rmo/Rmm.

**L273: agreed could this be due to mean bias for example?**

As mean is related with variance, yes, it is a possibility. mean bias could contribute. we will add a check for mean bias in the revised analysis and discuss it.

**L302: bad not band**

The typo will be corrected to "bad".

We believe these changes will address your concerns and improve the manuscript's clarity, rigor, and scope. We will submit a revised version incorporating these revisions shortly. Thank you very much for the constructive comments.

Thanking you,

Yashas Shivamurthy

---

## Author Comment (AC3)

**Dear Dr. Neale**

I have carefully reviewed their response and am deeply disappointed. Not only did their reply fail to address the key points, but more importantly, the authors appear to be responding to non-existent arguments (as noted in my comments below). I believe this study is methodologically flawed and conceptually confusing. Therefore, I am convinced that this work should not be published.

Below is my response to their replies. The red text represents their original comments, the bold black text is my previous feedback, and the blue text contains my current responses.

**Reply:** Our latest responses are provided in green (earlier replies in red). We respectfully disagree with the comment referring to our argument as "non-existent." Unfortunately, some of the reviewer's remarks appear to be misleading, and it seems that the reviewer has been inclined toward rejecting the manuscript from the outset, which we believe is unfair.

**Comment-1:** The article only explains how signal and noise variance are defined and calculated. Since variance itself is not the actual component, it is unclear how the signal and noise are extracted from the data. The concept and defintion are totally different between the variance and the variable itself.

**Reply:** There are numerous paper on how signal and noise components are extracted from model data and some of them are cited here (e.g. Kang and Shukla, 2006; Scaife et al., 2014; Saha et al., 2016a; Scaife and Smith, 2018; Weisheimer et al., 2018 and many more). While inter-ensemble spread is considered as noise/internal component, the ensemble mean is the signal/external component (equation 1 and 2 respectively in our manuscript). How signal and noise are extracted from data is clearly mentioned in lines 104-108 of the manuscript, section 2.3.1.

**Comment:** The articles cited by the authors only discuss signal variance and noise variance. It is problematic to treat the ensemble mean directly as the signal/external component. As a measure of variability, the signal should not be constrained by sign—how does one interpret a "positive signal" versus a "negative signal"? Therefore, it is more appropriate to use the square of the ensemble mean to represent the signal.

In the author's statements in lines 104–109 as below, I could not find a clear definition of either the signal or the noise.

respectively. Here, predictable and unpredictable components are termed external/signal and internal/noise components, respectively. The ratio of external to internal variance is known as the signal-to-noise ratio (SNR). If x is the precipitation field of the model, i is the year of the model integration (total year 'N'), and j is the number of ensemble simulations (total ensemble n = 52), then internal variance following Rowell et al. (1995), can be expressed as

**Reply:** The ensemble mean is always treated as the more reliable predictor; it is not a matter of a "positive" versus "negative" signal. This interpretation is incorrect. In our analysis, the *signal* (or external variance) represents the seasonal anomaly in the ensemble mean, with a correction term as defined in Equation (3). Our approach follows the methodology established by *Rowell et al.* (1995) and subsequently applied in numerous studies (e.g., Kang & Shukla, 2006; Scaife et al., 2014; Saha et al., 2016a; Scaife & Smith, 2018; Weisheimer et al., 2018). To avoid confusion, we have included excerpts below, from several of these references. Regarding the suggestion to use the "square of the ensemble mean" to represent the signal, we are not aware of any such approach in the existing literature. We have followed the standard and widely accepted method for estimating signal and noise components, as described in our manuscript (beginning at line 104).

We regret that the reviewer was unable to locate the definitions of *signal* and *noise* in the manuscript, which were clearly stated and also mentioned in our earlier response. Here is the cut-pest of our manuscript, describing signal and noise (beginning line 104).

**2.3.1 ANalysis Of VAriance (ANOVA) method**

In this method, the total variance is split into signal and noise components, i.e. external ( $\sigma_{EXV}^2$ ) and internal ( $\sigma_{IV}^2$ ) variances, respectively. Here, predictable and unpredictable components are termed external/signal and internal/noise components, respectively. The ratio of external to internal variance is known as the signal-to-noise ratio (SNR). If x is the precipitation field of the model, i is the year of the model integration (total year 'N'), and j is the number of ensemble simulations (total ensemble n = 52), then internal variance following Rowell et al. (1995), can be expressed as

$$\sigma_{IV}^2 = \frac{1}{N(n-1)} \sum_{j=1}^n \sum_{i=1}^N (x_{ij} - \overline{x_i})^2$$
 (1)

where  $\overline{x_i} = \frac{1}{n} \sum_{j=1}^{n} x_{ij}$  is the ensemble mean of the model for a year and the degrees of freedom is N(n – 1). The variance of ensemble mean  $(\sigma_{EV}^2)$  can be estimated as

$$\sigma_{EV}^2 = \frac{1}{(N-1)} \sum_{i=1}^{N} (\overline{x_i} - \overline{\overline{x}})^2 \tag{2}$$

where  $\overline{\overline{x}} = \frac{1}{Nn} \sum_{j=1}^{n} \sum_{i=1}^{N} x_{ij}$  is the average over all year and all ensemble. However, the variance of the ensemble mean is a biased estimate of external variance (Scheffe, 1959). As the number of ensemble members is not very large (here 52), the ensemble mean contains residual internal variability. Therefore, the external variance may be estimated following Scheffe

**Paper by Rowell et al. (1995), describe how signal/variability due to SST/external and noise/internal components, which are based on 'analysis of variance' methodology, are calculated.**

**RAINFALL VARIABILITY OVER NORTH AFRICA**

The next stage of research will be to explore the physical mechanisms which link the SST patterns to seasonal rainfall variability. Circulation changes over north Africa will be examined in a later publication, and some global-scale circulation patterns associated with Sahelian rainfall anomalies are presented by Ward et al. (1994).

Given that SST patterns are often predictable at least a few months in advance, this offers hope for the production of skilful forecasts of seasonal JAS rainfall anomalies averaged over the Sahel, Soudan and Guinea Coast. Indeed, such forecasts have now been issued by the UK Meteorological Office for the Sahel region since 1986, and for the Soudan and Guinea Coast regions since 1992, on an experimental basis (see Ward et al. (1993) for details). In order that such forecasts achieve maximum utility, further research is required on the variations of rainfall—SST relationships within the large regions used here and within the July to September season.

**ACKNOWLEDGEMENTS**

We are very grateful for the hard work of John Owen who set up and monitored many of the GCM experiments described here, and to David Parker who was involved in the early part of this work. We are also indebted to Mike Hulme (Climatic Research Unit, Norwich), who provided us with the observed rainfall data on the same grid as the GCM (under UK DoE contract PECD 7/12/78), and to John Rowell, whose statistical advice led to a much improved appendix.

**APPENDIX**

Here we reproduce the statistics required to separate the estimated total variance of simulated rainfall amounts (or any other parameter)  $(\sigma_{TOI}^2)$ , into an SST-forced component  $(\sigma_{SST}^2)$  and an internal variability component  $(\sigma_{TNI}^2)$ . Although in this paper the technique is specifically applied to tropical north African seasonal and monthly rainfall totals, it will also prove useful as a general analysis tool for understanding natural climate variability and potential atmospheric predictability on any chosen time- or space-scale.

We consider a generalized case of N ensembles (years of SST forcing), each with n members. The simulated rainfall amounts for each experiment are 'modelled' as the sum of two independent components:

$$x_{ij} = \mu_i + \varepsilon_{ij} \tag{A.1}$$

where:  $x_{ij}$  = simulated rainfall, i = 1, ..., N is associated with a particular year (i.e. an ensemble of experiments with the same SST forcing), j = 1, ..., n is associated with a particular member of the ensemble (i.e. identifies the initial atmospheric conditions),  $\mu_i$  = component of rainfall due to SST forcing, and  $\varepsilon_{ij}$  = anomalous rainfall due to internal variability

'analysis of variance' methodology, using a so-called 'n ndom-effects' model (see, for ndom-effects' model (see, for ndom-effects') model (see, for ndom-effects) model (see, for ndom-eff

First, the internal variability is easily estimated by computing the variance of each datum's deviation from its ensemble mean:

$$\hat{\sigma}_{\text{INT}}^2 = \frac{1}{N(n-1)} \sum_{i=1}^{N} \sum_{j=1}^{n} (x_{ij} - \overline{x_i})^2 \qquad (N(n-1) \text{ being the degrees of freedom})$$
 (A.2)

700

D. P. ROWELL et al.

where  $\hat{y}$  denotes an estimate of y, and:

$$\overline{x_i} = \frac{1}{n} \sum_{j=1}^{n} x_{ij}$$
 (the ensemble mean for the *i*th year). (A.3)

In order to estimate the variability due to SST forcing, we must first estimate the variance of the ensemble means:

$$\hat{\sigma}_{EM}^2 = \frac{1}{N-1} \sum_{i=1}^{N} (\overline{x_i} - \overline{x})^2 \qquad (N-1 \text{ being the degrees of freedom}) \qquad (A.4)$$

where:

$$\overline{x} = \frac{1}{nN} \sum_{i=1}^{N} \sum_{j=1}^{n} x_{ij}$$
 (the mean of all data). (A.5)

Now, using a standard result from 'analysis of variance', we can also express  $\sigma_{\rm EM}^2$  in terms of  $\sigma_{\rm EM}^2$  and  $\sigma_{\rm SST}^2$  (e.g. Scheffe 1959, p. 226):

$$\sigma_{EM}^2 = \sigma_{SST}^2 + \frac{1}{n} \sigma_{INT}^2. \qquad (A.6)$$

This essentially states that the variance of the ensemble means is a biased estimate of the variance due to SST forcing. This is because with finite n each ensemble mean  $(x_i)$  still contains an element of internal variability (each ensemble mean is only an estimate of  $\mu_n$  not equal to  $\mu_n$ ), so that  $\sigma_{FN}^2$  overestimates  $\sigma_{SST}^2$ .

estimate of  $\mu_i$ , not equal to  $\mu_i$ ), so that  $\sigma_{\rm EM}^2$  overestimates  $\sigma_{\rm SST}^2$ . Thus, from Eqs. (A.6), (A.4) and (A.2), the variance due to SST forcing may be estimated as:

$$\hat{\sigma}_{SST}^2 = \hat{\sigma}_{EM}^2 - \frac{1}{n} \hat{\sigma}_{INT}^2 = \frac{1}{N-1} \sum_{i=1}^{N} (\overline{x}_i - \overline{x})^2 - \frac{1}{Nn(n-1)} \sum_{i=1}^{N} \sum_{i=1}^{n} (x_{ij} - \overline{x}_i)^2$$
(A.7)

[see note (i)]

This then leads to an estimate of the total variance:

$$\hat{\sigma}_{TOT}^2 = \hat{\sigma}_{INT}^2 + \hat{\sigma}_{SST}^2 \qquad (A.8)$$

[see note (ii)],

and finally, the ratios of components of variance:

$$\frac{\hat{\sigma}_{\text{SST}}^2}{\hat{\sigma}_{\text{TOT}}^2}$$
 and  $\frac{\hat{\sigma}_{\text{INT}}^2}{\hat{\sigma}_{\text{TOT}}^2}$

Notes:

- (i) Because of the subtracted term in Eq. (A.7), the distribution of  $\sigma_{\text{SST}}^2$  includes a few negative values, which are sometimes produced by chance. However, this is only likely when n and N are small and when  $\sigma_{\text{SST}}^2/\sigma_{\text{TOT}}^2$  is small. Such negative values are best reset to zero, but in any case do not occur in the results presented here.
- (ii) It is perhaps tempting to estimate  $\sigma_{\text{TOT}}^2$  as:  $\frac{1}{Nn-1}\sum_{i=1}^{N}\sum_{j=1}^{n}(x_i-\overline{x})^2$ . However, this is a *biased* estimate, because it fails to account for the impact that the

However, this is a biased estimate, because it fails to account for the impact that the make-up of the data has on the number of degrees of freedom, i.e. that the data has two components of variability. The total variance is correctly estimated only by the sum of its two components (Eq. (A.8)).

Where equation (A.2) is equation (1) in our manuscript (A.4) is equation (2) in our manuscript external variance/SST forced variance (A.7) is equation (3) in our manuscript. Below is from Kang and Shukla (2006), *Dynamic Seasonal Prediction and Predictability of the Monsoon, in: The Asian Monsoon, edited by Wang, B pp. 585–612, Praxis, Springer, Berlin.*

**Book Chapter by Kang and Shukla (2006)**

Figure 15.1. Variances of summer mean precipitation anomalies for the 21-year period (1979-1999). (a) CMAP observation precipitation, (b) JMA, (c) KMA, (d) NASA, (e) NCEP, and (f) SNU prediction models. The variance of each model is computed using all the ensemble members of the 21-year predictions. The contour interval is 1, 3, 6, 12, 24, and 48 mm² day² and light and dark shadings indicate a variance of more than 3 and 12 mm² day², respectively.

regions, particularly over the Asian monsoon region. The difference among the model variances is partly related to the difference in the mean climatology and to the different combinations of model physics. But, it is difficult to identify the model physics responsible for generating such large differences.

The total variance  $(\sigma_{TOT}^2)$  is divided into the external  $(\sigma_{SST}^2)$  and internal variances  $(\sigma_{TNR}^2)$ ; Rowell, 1998). The ensemble mean is considered to be the external component of the prediction forced by the SST forcing, and the deviation from the ensemble mean is the stochastic internal component of the prediction. The

Sec. 15.3]

Limit of seasonal predictability 593

internal variance can then be expressed as:

$$\sigma_{INR}^2 = \frac{1}{N(n-1)} \sum_{i=1}^{N} \sum_{j=1}^{n} (x_{ij} - \overline{x_i})^2$$
 (15.1)

where x is the precipitation, i indicates the individual year, N = 21, j is the ensemble member, and n is 6 to 10 for different models.  $\bar{x}_i$  is the ensemble mean. The external variance is obtained by the mean square of the deviation of each year's ensemble mean from the climatological mean and with a consideration of bias correction, as in Rowell (1998):

$$\sigma_{SST}^2 = \sigma_{EN}^2 - \frac{1}{n} \sigma_{INR}^2$$
 and  $\sigma_{EN}^2 = \frac{1}{N-1} \sum_{i=1}^{N} (\overline{x_i} - \overline{\overline{x}})^2$  (15.2)

where  $\overline{x}$  is the climatological mean and  $\overline{x} = 1/(Nn) \sum_{i=1}^{N} \sum_{j=1}^{n} x_{ij}$ . It should be noted that the sum of external and internal variances expressed above is equal to the total variance.

Figure 15.2(a-e) show the external variances of various models, and Figure 15.2(f-j) the internal variances. The signal-to-noise ratio, the ratio of the external part to the internal part of the corresponding model, is shown in Figure 15.2(k-o). All the models produce large external variances over the tropical oceans that are much larger than the internal variance of the same model, particularly the ENSO region. This result indicates that tropical rainfall is less controlled by atmospheric internal processes and is thus less predictable for a given SST condition. In the extratropics, on the other hand, the internal variances are bigger than the external variances of the same model (Figures 15.2(k-o)), and therefore the extratropical atmosphere is more controlled by non-linear stochastic processes and less

**Comment 2:** The article tries to discuss and analyze the paradox, but the purpose of using Nino3.4 to predict precipitation remains unclear. What is the intention behind comparing it with dynamic models? Is it to demonstrate whether the actual or potential forecast skill of dynamic models is higher or lower, reasonable or unreasonable? The objective is not clearly stated. Moreover, can using Nino3.4 to predict precipitation effectively achieve these goals? Would the forecast skill be reliable? Was the forecast skill mentioned in the article derived from training or test data? Similarly, were other modes affecting precipitation in the Indian region, such as IOD, considered?

**Reply:** The idea is to asses prediction skill of not only predictants (i.e. ISMR, PACR), but also the fidelity in simulating global predictors (e.g. ENSO) and their teleconnections. Figure 9 shows multiple correlations involving major global predictors (Niño3.4, IOD, PDO, AMO) and sub-seasonal components.

**Comment:** Your response does not address my question. Such a simple linear regression approach is unreliable and insufficient to explain any core issues discussed in this paper.

**Reply:** We would like to emphasize that *linear regression* is a well-established and reliable statistical tool, particularly when the relationship is found to be statistically significant. Some of the questions raised earlier (in black text above), such as "*Was the forecast skill mentioned in the article derived from training or test data?*", are not relevant in this context and would not typically arise from a domain expert. Our analysis is based on coupled model re-forecast data; therefore, the concepts of "training" and "test" datasets do not apply here.

**Comment 3:** Rowell (1995) never defined signal variance and noise variance using ANOVA. While they did mention ANOVA, it was only used for statistical testing. The authors should revisit Rowell (1995) to better understand the content. ANOVA has exactly defintion in statistics, which should be followed to avoide unnecessary confusion.

**Reply:** Please look into page no 699 of Rowell et al. (1995). https://rmets.onlinelibrary.wiley.com/doi/epdf/10.1002/qj.49712152311

,which mention "The approach we use to estimate the components of variance closely follows an 'analysis of variance' methodology ..."

**Comment:** I could not find the answers provided by the authors in Rowell (page 699) as below. It should be noted that ANOVA has a rigorous statistical definition. The authors, however, only performed variance partitioning, not ANOVA.

**RAINFALL VARIABILITY OVER NORTH AFRICA**

The next stage of research will be to explore the physical mechanisms which link the SST patterns to seasonal rainfall variability. Circulation changes over north Africa will be examined in a later publication, and some global-scale circulation patterns associated with Sahelian rainfall anomalies are presented by Ward et al. (1994).

Given that SST patterns are often predictable at least a few months in advance, this offers hope for the production of skilful forecasts of seasonal JAS rainfall anomalies averaged over the Sahel, Soudan and Guinea Coast. Indeed, such forecasts have now been issued by the UK Meteorological Office for the Sahel region since 1986, and for the Soudan and Guinea Coast regions since 1992, on an experimental basis (see Ward et al. (1993) for details). In order that such forecasts achieve maximum utility, further research is required on the variations of rainfall–SST relationships within the large regions used here and within the July to September season.

**Reply:** This comment appears to be misleading. The reviewer should refer to the entire page 699 of *Rowell et al.* (1995) rather than quoting an isolated paragraph, which risks misrepresenting the context. To clarify, we have now included pages 699 and 700 from *Rowell et al.* (1995) in our response to Comment 1. As clearly indicated in the highlighted section (marked with a red rectangle), the method indeed follows the "analysis of variance" (ANOVA) approach. Therefore, the argument questioning whether it is ANOVA-based is not meaningful.

**Comment 4:** I do not understand the meaning of the statement: "The use of the orthogonality assumption is a methodological simplification to partition variance across time scales; it does not imply the absence of physical co-variability." Do physical and mathematical co-variability have different interpretations? In my opinion, if two quantities are physically related, they cannot be assumed to be orthogonal in mathematics. Additionally, I do not comprehend the authors' claim that "sub-seasonal components are the building blocks of the seasonal mean." Following this logic, all time scales would be sources of error, since hourly components are the building blocks of the daily mean, and daily components are the building blocks of the weekly mean, and so on.

**Reply:** The argument why we are using assumption of orthogonality and not the actual one, lies on the fact that it is challenging (if not impossible) in a non-linear system to separate individual components.

Sub-seasonal components of the monsoon particularly have clear preferred band. Some of the band are more vigorous in terms of their spatial scale, strength than the others. In terms of their contribution to the mean and variability/predictability also varies. While MISOs have very large spatial structure and strong sub-seasonal variability, their contribution to year-to-year monsoon rainfall variability is minimum (weak negative correlation). So, clearly, we are not talking here about hourly/daily events but some known and prominent sub-seasonal variability/bands, which shape the seasonal monsoon rainfall of a year. Here are literatures, cited in support of our arguments (Saha et al., 2019; Borah et al., 2020). Some important papers in the similar lines but not cited here are.

**Comment:** It seems no basis to argue the "challenging to separate" as a justification for such an assumption. This is the most critical weakness of the study: on the one hand, it attempts to examine the effect of A on B using linear statistical analysis, while on the other hand, it assumes that A and B are orthogonal, implying that their covariance (or correlation coefficient) is zero.

I am drawing this inference based on the authors' own argument. You may choose to ignore or omit other scales of the atmospheric process, but I cannot overlook them. Isn't that ?

**Reply:** We thank for your comment. The assumption of orthogonality is a widely used and mathematically consistent methodological simplification in the analysis of complex, multiscale geophysical systems. It does not imply that the underlying physical processes are truly independent, but rather provides a tractable framework to *partition total variance* among distinct temporal (or spatial) scales. In nonlinear climate systems, exact separation of

variability across scales is not possible because physical processes are dynamically coupled and often nonlinearly interacting. However, for the purpose of statistical decomposition and diagnostic analysis, an orthogonal representation allows us to quantify the *relative contributions* of different time-scale components (e.g., sub-seasonal, interannual, decadal) to the total variance, without double-counting shared variability.

This approach is conceptually similar to other well-established methods that rely on orthogonality, such as Empirical Orthogonal Function (EOF) analysis, spectral decomposition, and ANOVA, where the basis functions or components are constructed to be orthogonal in the statistical sense, even though the corresponding physical processes may interact. In this context, orthogonality is a mathematical convenience, not a physical claim of independence.

Thanks for your comment regarding "sub-seasonal components are the building blocks of the seasonal mean.". We will add text here to make thing clearer to the reader.

**Comment 5:** So I have to feel sorry to decline this work again. The topic is interesting that is the reason why I agreed with reviewing it. Unfortunately I do not learn more from this work. To my understanding, the paradox should be from the "defintion" of potential predictability. The ratio of signal to noise may not well represent the potential predictability. If authors wish to work this problem, I suggest them to seek other measures to quantify the potential predictability.

**Reply:** We wish, if you could have read the full manuscript. The main content of the manuscript is the following:

- i) Perfect model framework is used to estimate potential predictability of seasonal anomaly, which often shows paradoxical behaviour. 'Analysis of variance' framework is used for calculating 'signal' and 'noise' components using 52-ensemble member re-forecasts.
- ii) Here we argue that 'perfect model framework' is not adequate, as the error growth is not from only initial condition errors but also from other sources, like physics, numerical scheme etc. We demonstrated that sub-seasonal component, which is part of the physics, adds error (biased contribution) in the seasonal forecast anomaly (i.e. Figure 7). However, 'perfect model framework' assumes, ensemble spread solely attributed to initial condition error. Consequently, true limit of predictability is not known. So, here our argument matches with your point of view that the method of estimating PPL based on perfect model framework is inadequate. We have already mentioned it in lines 337-344, in the last para of section 3.3

**Comment :** The authors appear to lack a clear understanding of the PPL issue. PPL is fundamentally a product of the "perfect model" framework. Once model errors are taken into account, it ceases to be a PPL problem. Therefore, the very premise of this study is conceptually inconsistent.

iii) Finally we propose a method for estimating PPL, which is free from paradox (section 3.4). Therefore, we believe the rationale provided for rejection does not fully capture the merits of the manuscrip

**Reply:** We respectfully disagree with the reviewer's interpretation. The PPL is indeed defined within the *perfect-model* framework; however, the objective of this study is to evaluate its practical limitations. We do not attempt to redefine PPL, but rather to show that the conventional signal-to-noise—based estimation becomes biased when ensemble spread is influenced not only by initial condition errors but also by internal model processes such as physics and sub-seasonal variability. In this sense, our work extends, not contradicts, the original concept by identifying how real-world model imperfections distort the theoretical upper bound of predictability. This perspective is consistent with earlier studies (e.g., Kumar & Hoerling, 2000; Scaife et al., 2014; Weisheimer et al., 2018) that recognized limitations in the perfect-model assumption and gave plausible reasons. Therefore, the premise of our study is conceptually coherent and offers a constructive refinement to the understanding of PPL.